# HashMark: Watermarking Tabular/Synthetic Data For Machine Learning Via Cryptographic Hash Functions

## Abstract

As enterprises increasingly rely on data for decision-making and machine learning pipelines, ensuring data provenance, ownership, and responsible use has become essential. Data watermarking offers a promising solution by embedding imperceptible markers into datasets, enabling traceability and accountability. While prior work has primarily focused on perceptual domains such as images, audio, and text, watermarking for tabular data remains underexplored despite its central role in enterprise systems. Tabular data presents unique challenges due to its heterogeneity, lack of redundancy, and susceptibility to structural modifications.

We introduce HashMark, a suite of cryptographic watermarking protocols explicitly designed for tabular datasets. Our methods embed bits into table cells using seeded hash functions, achieving *data-type agnostic*, high-fidelity watermarking with minimal distortion. We present two complementary schemes: (i) $\text{HashMark}_1$, a sparse embedding mechanism that modifies only $\Theta(1)$ cells, and (ii) $\text{HashMark}_2$, a dense embedding mechanism that enforces uniform statistical properties across the dataset while supporting categorical and alphanumeric domains. Both schemes feature low detection cost, broad applicability, and formal fidelity guarantees.

Extensive experiments across various settings demonstrate that HashMark maintains downstream model performance while significantly improving the quality of the watermarking scheme, when compared to prior work. Our results establish hash-based watermarking as a simple, efficient, and general solution for securing tabular data against unauthorized use, while also enabling scalable data governance.

## 1 Introduction

As data-driven applications grow in significance, ensuring data integrity, provenance, and ownership is increasingly critical. Data watermarking—the practice of embedding imperceptible markers into datasets—has emerged as a valuable tool for protecting intellectual property, preventing unauthorized use, and verifying authenticity. This is especially relevant when data is shared, sold, or used to train machine learning models, as it provides mechanisms for tracing data lineage and safeguarding against misuse. With the rise of generative models and synthetic data, watermarking also ensures traceability of AI-generated content.

**Prior Work** Previous research on watermarking has largely focused on image, audio, or text data (Ahmadi et al., 2021; Tan et al., 2023; Yamni et al., 2022; Zhang et al., 2022; Zhong et al., 2021), while tabular data—one of the most common formats in machine learning—has received less attention. Watermarking tabular data is challenging due to (i) the lack of perceptual redundancy, where small changes can be impactful, (ii) mixed data types requiring tailored strategies, and (iii) the need for resilience against insertions, deletions, and foreign key modifications. Existing tabular watermarking methods (Agrawal & Kiernan, 2002; Hu et al., 2018; Hwang et al., 2020; Kamran et al., 2013; Li et al., 2022; Lin et al., 2021; Shehab et al., 2008; Sion et al., 2003) often focus on relational databases and either modify specific data points or embed statistical identifiers. More recent approaches (He et al., 2024; Zheng et al., 2024; Ngo et al., 2024) have targeted general tabular data, yet challenges remain regarding computational complexity, scalability, and storage requirements. More information pertaining to related work is deferred to Section A.

Table 1: Comparison of HashMark with prior works (transposed). Detection Cost refers to the information needed to detect the watermark efficiently. "# Modification" refers to the number of cells that need to be modified to embed the watermark.

|  | Ngo et al. | Zheng et al. | HashMark$_1$ | HashMark$_2$ |
|---|---|---|---|---|
| # Modification | All | All | $\Theta(1)$ | All |
| Fidelity | High | High | Very High | High |
| Deletions | Allowed | Allowed | Limited | **Allowed** |
| Permutations | Allowed | Allowed | Limited | **Allowed** |
| Data Types | Numerical | Any* | **Any** | Any |
| Detection Cost | High | Very High | **Very Low** | Low |

**Our Motivation**   We focus on watermarking in *non-adversarial enterprise settings*, where data flows across multiple departments and systems. In such contexts, employees typically do not attempt to remove watermarks, which enables effective tracking of data lineage, ensures integrity, and facilitates compliance with internal policies and regulations. Embedded markers enable organizations to monitor data movement, quickly identify discrepancies, and maintain accountability throughout the data lifecycle.

The rise of synthetic data further motivates the use of watermarking, as organizations must distinguish between synthetic datasets and originals while preserving both privacy and utility. While no watermarking scheme is entirely immune to removal (Zhang et al., 2024b), its practical value lies in raising the cost of misuse and enabling accountability. Our work enhances the applicability of tabular data, thereby strengthening enterprise data governance in realistic, non-adversarial scenarios.

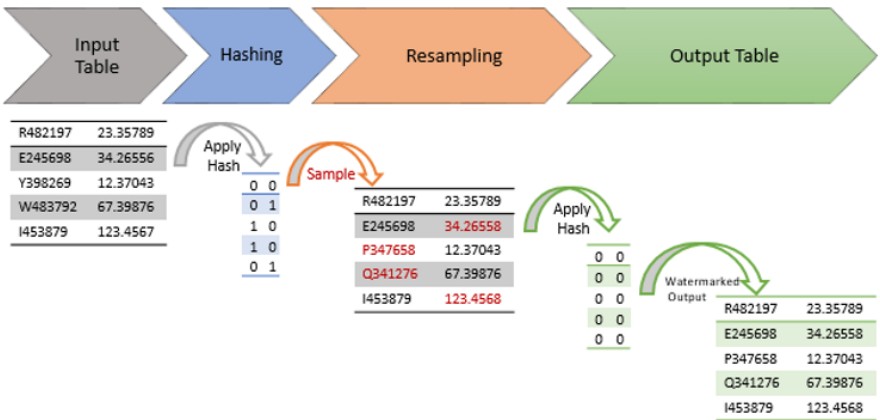

Figure 1: HashMark$_2$: On the left is the source input table, to be watermarked, containing cells of two columns - one text and the other numerical. After applying the hash function to each cell, the hashed values are shown next. In the middle, we show how values are adjusted to hash to 0. For text data, we replace it with a new value, and for numerical data, we add in the smallest decimal place. On the right is the watermark embedded table where all cells hash to 0.

## 1.1 OUR CONTRIBUTIONS

We introduce HashMark, a suite of simple yet powerful watermarking protocols for tabular datasets. Our approach embeds bits into selected table cells using a *cryptographic, seeded hash function*, ensuring that the output looks uniformly random without the knowledge of the seed. A hash function is versatile in its agnosticism regarding the input data type, working with both numeric and alphanumeric inputs.

We present two variants, HashMark$_1$ and HashMark$_2$, each offering unique properties. In both schemes, we map cell contents to a target bit (0 or 1) via the seeded hash function. If the cell content does not map to the target bit, we carefully modify the cell values while preserving the dataset's fidelity. For numerical values, we make minimal perturbations (e.g., incrementing by $10^{-c}$). For

alphanumeric values, we apply rejection sampling from the original distribution. The rejection sampling technique can also be extended to numerical values, as we describe later.

**HashMark$_1$.** For static datasets (e.g., unique IDs, timestamps, categorical labels), HashMark$_1$ modifies only a constant $\ell \ll N$ cells, ensuring high fidelity. HashMark$_1$ employs two pseudorandom generators (PRGs). A PRG uses a seed, ensuring that the output appears random without knowledge of this seed. We use the first PRG $G_1$ to derive $\ell$ bits. We then use the second PRG $G_2$ to identify the dataset's $\ell$ cell locations. Each chosen cell is adjusted until it hashes to the selected bit. Here, $\ell \ll N$ where $N$ is the number of cells in the datasets, which guarantees very high fidelity. For detection, we first use $G_2$ to identify the $\ell$ cell locations. Then, employing the hash function, we retrieve the bits embedded at these positions. Finally, using $G_1$, we verify whether the retrieved bits match the embedded information. HashMark$_1$ requires the knowledge of the seeds for watermark detection. Note that the security and correctness of its detection algorithm stem from the security of the underlying cryptographic constructions. Minor permutations or deletions of rows compromise detection since they disrupt cell positioning. On the other hand, if permutations aren't allowed, then removing the watermark is difficult as the embedding locations are pseudorandom.

**HashMark$_2$.** Figure 1 pictorially represents HashMark$_2$, where the same target bit (say 0) is embedded in *all* (or, $O(N)$ cells as relaxed later). It uses the hash function for the binary mapping and then applies the above-outlined "adjustment" procedure to ensure that every cell maps to 0 under the seeded hash function. This approach, though appearing similar to the red-green paradigm of Ngo et al. (2024) and Zheng et al. (2024), is vastly simpler and more secure. Indeed, while Ngo et al. (2024) relied on an insecure seed for mapping to red or green, Zheng et al. (2024) required the source dataset for detection. Instead, our detection algorithm relies on a statistical test, and the embedding algorithm can be instantiated with several approaches, such as perturbing the values by adding $10^{-c}$ for some constant $c$ or simply rejection sampling until a certain threshold number of the entries in a row map to the desired bit. Looking ahead, we employ both techniques for numerical values in the experiments section.

However, critically, our reliance on the seeded hash function ensures that it supports any data type (a feature missing from the work of Ngo et al. (2024)) and does not require the source dataset to identify the watermarking (a feature of Zheng et al. (2024)). Like HashMark$_1$, we adjust the cell content to obtain the mapping to the target bit of 0, at every cell position. Unlike HashMark$_1$, HashMark$_2$ will rely on a statistical test to determine if the dataset was watermarked.

In conclusion, our suite of protocols HashMark satisfies:

- **High Fidelity**: The dataset changes are minimal when values are perturbed by adding $10^{-c}$, and nonexistent when using rejection sampling, since samples are drawn from the same distribution.
- **Low Detection Cost**: Detection in HashMark requires only the hash (and PRG for HashMark$_1$) seeds due to its simpler design, whereas Ngo et al. (2024) needs column pairings and Zheng et al. (2024) requires the full source dataset.
- **Support for Any Data Type**: HashMark can *support any data*, as explained above. In contrast, Ngo et al. (2024) cannot handle categorical data, and although Zheng et al. (2024) claims broad support, it is unclear how their method applies to textual data.[1] Hence, in Table 1, we mark their support as Any$^*$.

## 2 PRELIMINARIES

**Notations.** For $n \in \mathbb{N}^+$, we denote by $[n]$ the set $\{1, \ldots, n\}$. For a set $X$, we denote by $x \xleftarrow{\$} X$ that a value $x$ is sampled uniformly at random from $X$.

---

[1]Zheng et al. (2024) focuses on categorical data (e.g., education level, marital status). Their watermarking distorts integer distributions by adding floating-point perturbations, which harms utility. Restricting to integer perturbations could leave gaps in the column range, so we argue that such columns should not be watermarked. Moreover, they neither support unrestricted alphanumeric data (e.g., ASINs) nor evaluate such cases.

**Seeded Hash Function.** A function $\mathcal{H} : \mathcal{S} \times \mathcal{X} \to \mathcal{Y}$ is a hash function, modeled as a random oracle, if the computation of $\mathcal{H}(S, X)$ for a random $S \xleftarrow{\$} \mathcal{S}$ and any $X \in \mathcal{X}$ is indistinguishable from $Y \xleftarrow{\$} \mathcal{Y}$. In our application, we will suppress the presence of the seed distribution $\mathcal{S}$ and we will set $\mathcal{Y} := \{0, 1\}$.

## 3 Problem Formulation

Our dataset is a matrix $\mathbf{X} \in \mathbb{R}^{m \times n}$ containing numerical, alphabetical, and alphanumeric values. The goal is to construct a watermarked dataset $\mathbf{X}_w$ with the following properties:

**Fidelity:** $\mathbf{X}_w$ remains close to $\mathbf{X}$. For numerical data, we show closeness in $L_\infty$ distance (Theorem 1) and for categorical data, we show closeness by Jensen-Shannon Divergence (Theorem 2).

**Detectability:** The watermark can be efficiently and reliably detected—cryptographically in the first variant, and statistically in the second.

**Robustness:** $\mathbf{X}_w$ withstands common perturbations, such as row/column removal, permutations, and cell modifications.

**Utility:** $\mathbf{X}_w$ supports downstream tasks (e.g., machine learning) with negligible accuracy loss, as confirmed empirically.

## 4 HashMark: Element Wise Tabular Watermarking

At its core, any watermarking approach needs to ensure that the utility of the data is preserved even after embedding the watermark. Furthermore, the detectability of the watermark is preserved even after modification by both adversarial and honest actions. We have two constructions $\mathsf{HashMark}_1, \mathsf{HashMark}_2$ with various properties and an implicit trade-off.

However, before examining the constructions, it is instructive to consider the commonalities. Both the constructions will rely on applying a seeded hash function $\mathcal{H}$ that can take any inputs and produce an output bit. Such a binary hash function enables us to map any cell (numerical, textual, categorical, etc.) to either 0 or 1, depending on the function's description. They will also rely on modifying a cell's contents through invoking the function $\mathsf{Generate}$ (until it satisfies some $\mathcal{H}$-based property). The question remains of how to instantiate this function.

### 4.1 Defining Generate

The crux of our construction lies in instantiating the function $\mathsf{Generate}$, which modifies dataset content to satisfy the hashing requirement. In this section, we define this function and present some optimizations. Before proceeding, we would like to highlight an essential caveat in our approach to watermarking. Consider a column $C$ with a fixed range (e.g., marital status, education level, designation, or salary tiers). Applying a hash function that maps values to bits (0 or 1) can force certain elements to hash to an undesired bit. This would remove those elements from the range in the watermarked dataset, skewing the distribution and harming utility. Note that Zheng et al. (2024) suggests embedding in these columns by first mapping these entries into distinct integers and then reverting to their numerical-based approach. However, this skews the distribution and can harm correlations. To avoid this, we do not embed watermarks in such columns; instead, we treat every element in the range as "valid," i.e., as hashing to the desired bit.

In the ensuing discussion, we focus solely on generating values for the remaining attributes/columns. We will focus on embedding the watermark and later define fidelity, i.e., how closely the watermarked distribution resembles the un-watermarked one. The proofs of the following are deferred to Section E in the appendix.

**Numerical Values.** Suppose a column $C$ consists of numerical data, specifically floating-point values. In that case, the generate function can take the old value and add $10^{-c}$ for some constant $c$ that is a scheme parameter. This ensures that the perturbation does not adversely impact the fidelity. Formally, we have the following theoretical guarantee, as measured by the expected difference in $L_\infty$ between the unwatermarked and watermarked distributions.

**Theorem 1.** *Let* $\mathbf{X}$ *be the original dataset and* $\mathbf{X}_w$ *be the watermarked dataset of size* $N$ *where* $x_i' \in \mathbf{X}_w$ *is generated as follows:*

$$x_i' = x_i + k_i \cdot 10^{-c},$$

*where* $k_i = \min\{k \geq 0 \mid \mathcal{H}(x_i + k \cdot 10^{-c}) = 0$, $\mathcal{H}$ *is a seeded hash function as defined before, and* $c \geq 0$ *is some integer. Then,*

$$\mathbb{E}[||\mathbf{X} - \mathbf{X}_w||_\infty] \leq (\ln N + 2) \cdot 10^{-c}$$

Our approach can be easily extended to support truncation up to $b$ decimal places if only the value until the first $b$ decimal places is included in the input to $\mathcal{H}$.

**Alphanumeric/Textual Data.** In the case of textual data, the generate function can reject and resample from the underlying distribution for the feature $\rho_i$. Then, one can measure the fidelity of the watermarked dataset by measuring the Jensen-Shannon Divergence (Lin, 1991) between the watermarked and the un-watermarked dataset. Formally, we get the following theoretical guarantee:

**Theorem 2.** *Let* $\rho$ *be the distribution of an alphanumeric column where we embed the watermark. Let* $\rho'$ *be the modified distribution consisting only of those values that hash to 0. Then, the Jensen-Shannon Divergence is:*

$$JSD(\rho||\rho') = \frac{3}{4}\log(\frac{4}{3}) \approx 0.215$$

**Preserving Correlations.** Datasets often contain correlations between various features or attributes. Any watermarking approach should ensure that these correlations are preserved. Rejection sampling column-wise can often lead to a loss of such correlations. We now detail how to preserve correlations.

- Let $\rho$ be a probability distribution that defines the underlying dataset. This can contain both categorical (aka alphanumeric values) and numerical values. For example, a synthetic data generation algorithm (such as the ones employed in our experiments) is trained on a source (i.e., original dataset), which yields such a distribution $\rho$ from which one can sample as many rows as needed. These synthetic data algorithms have been experimentally demonstrated to be closely aligned with the original dataset for various machine learning tasks, providing a heuristic proof of correlation preservation.

- Let $R \xleftarrow{\$} \rho$ be a row sampled from this distribution. Further, let this row $R$ be such that there exist cells that do not map to the desired bit.

- We can now reject $R$ and resample from $\rho$ until the sampled row satisfies the required constraint. However, such rejection and resampling until *every* cell maps to the desired bit can be computationally expensive. For $n$ columns, this can take $2^n$ time. Instead, one can choose a threshold $t$ such that if $t$ of the $n$ cells in a row $R$ map to the desired bit, it is marked as accepted. The detectability threshold can be suitably set to account for this modification.

The remainder of this section will focus on HashMark$_2$, deferring HashMark$_1$ to Section C. In brief, HashMark$_1$ is intended for static datasets where no modification of rows, columns, or their relative ordering is anticipated. Then, one can embed a pseudorandom number of bits in pseudorandom locations using a seeded hash function.

## 4.2 HashMark$_2$: Global Embedding

Unlike HashMark$_1$, HashMark$_2$ is more resilient to various perturbations and cell modification. The embedding approach is visually represented in Figure 1 and described in Algorithm 1. The crux of the strategy is to embed a global bit (say 0) in *every* cell of the dataset $\mathbf{X}$ using a binary hash function $\mathcal{H}$—consequently, a watermarked table to have more values that hash to 0 than an unwatermarked table. Detection is performed by using the secret description of the hash function to hash the data and count the number of cells that map to zero. Additional methods can allow the user to check only a subset of locations, making a slight skew more pronounced. This approach has the versatility of embedding a watermark in an existing dataset or generating a watermarked dataset at the source. The latter is a setting suitable for synthetic data.

**Detecting** HashMark$_2$. To detect HashMark$_2$, we use a one-proportion z-test (Fleiss et al., 2013), which is a statistical test used to determine whether the single sample rate, for example, the success rate in the number of entries that map to 0, is significantly different from a hypothesized population rate. We define the null hypothesis as:

$$H_0 : \text{Dataset } \mathbf{X} \text{ is not watermarked}$$

However, we note that if the null hypothesis holds, then so does a hypothesis $H_{0,i}$ : The $i$-th column is not watermarked also hold. This reduces the problem of rejecting $H_0$ to simply rejecting $H_{0,i}$ for each column $i$.

Let $T_i$ represent the number of elements in the $i$-th *value* column that hash to 0. Under the $i$-th null hypothesis, $H_{0,i}$ should follow the Bernoulli Distribution $B$ with probability $1/2$ as an ideal hash function $\mathcal{H}$ will output 0 or 1 with probability 1/2. Let $m$ be the total number of rows, i.e., $T_i \sim B(m, 1/2)$ for a sufficiently large number of rows $m$. By the Central Limit Theorem (CLT), for large $m$, we obtain that:

$$2\sqrt{m}\left(\frac{T_i}{m} - \frac{1}{2}\right) \sim \mathcal{N}(0, 1)$$

where $\mathcal{N}(0, 1)$ is the normal distribution. Thus, the test statistic for a one-proportion $z$-test is:

$$z = 2\sqrt{m}\left(\frac{T_i}{m} - \frac{1}{2}\right) \tag{1}$$

For each column, the detection algorithm computes a $z$-score by counting the number of values that hash to 0. To account for multiple hypothesis testing (e.g., five columns at $\alpha = 0.05$), per-column thresholds $\alpha_i$ are adjusted (e.g., $\alpha_i = 0.01$). If a column's $z$-score exceeds its threshold, the null hypothesis is rejected, indicating a watermark. Otherwise, no conclusion is made.

To prevent spoofing (where forgers combine valid watermarked datasets), we use a secret seed in the hash function (Algorithm 1). Each dataset's watermark uses a unique $seed$, making concatenated forgeries detectable as inconsistent.

**Robustness to Deletion, Permutation.** It is clear that the permutation of rows does not impact the count $T_i$. $H_{0,i}$ is evaluated for every column $i$. This implies that the permutation of the column from position $i$ to some $j$ will still have its corresponding null hypothesis $H_{0,j}$ and will be evaluated. Note that the detection algorithm performs multiple hypothesis tests simultaneously. Therefore, removing columns implies that one has to compute $\alpha_i$ as a function of $\alpha$ and the number of remaining columns. This guarantees robustness to column deletion. Removal of rows implies a smaller $m$. This results in an increase in the error in the CLT approximation. However, in practice, a rule of thumb for applying the Z-test has been $m > 50$ (Contributions, 2025). However, if $m < 50$, one could apply the Z-test on $H_0$ and not individual $H_{0,i}$.

Finally, as remarked before, one can also modify the application of $\mathcal{H}$ to ensure support for truncation.

## 4.3 ANALYSIS ON REMOVAL OF HashMark

Before we look at the mathematical analysis, we discuss the modes of attack to remove the watermark. The property of the ideal hash function $\mathcal{H}$ implies that the perturbation of a cell content initially mapping to 0 can flip to 1, with a probability of 0.5. Further, a secret seed (of the seeded hash function) implies that an adversary, without knowledge of this seed, cannot determine the actual mapping of the bit.

This section will study the effort required for the perturbation to remove the watermark. Specifically, an adversary can only modify $r$ cells by adding noise to them. We will analyze the expected number of $r$. Note that an adversary, adding noise to every cell in a column, can remove the watermark. This is true for every scheme (He et al., 2024; Ngo et al., 2024; Zheng et al., 2024). Experimentally, we present the results for comparison with Ngo et al. (2024) in Section 5.

In the analysis below, we assume there are a total of $M$ values. Of this, $N$ is the number of values that have the property of hashing to a desired bit. In HashMark$_1$, we have $N = \ell$ while $M = mn$. In HashMark$_2$, we have $N = M = m$ as described above. The proof of the following two results is deferred to Section E in the appendix.

---

**Algorithm 1** Embedding Algorithm

---

**Input:** Sampling Algorithm for Dataset $\mathcal{D}$ Generate
Secret Seed seed
Number of Rows: $\ell$
Associated Distribution: $\rho$
Column *column* of dataset $\mathbf{X}$
$seed \xleftarrow{\$} \mathcal{S}$ //$\mathcal{S}$ *is the seed space of the hash function.*

---

**for** $i = 1$ **to** $\ell$ **do**
  **while** $\mathcal{H}(seed, \mathcal{D}[i]) \neq 0$ **do**
    $new\_value \leftarrow$ Generate$(\rho, \mathcal{D}[i])$ //*Additional parameters could include t for threshold-constrained*
    *sampling.*
    $\mathcal{D}[i] \leftarrow new\_value$
  **end while**
**end for**

---

**Proposition 1.** *Given values $val_1, \ldots, val_M$. Then, the minimum number of values needed to ensure that the Z-score remains $\alpha$ is given by:* $\alpha \cdot \frac{\sqrt{M}}{2} + \frac{M}{2}$

**Theorem 3.** *Let $r$ be the number of cells an adversary can modify. This modification is done by sampling noises $\epsilon_1, \ldots, \epsilon_r \xleftarrow{\$} \mathcal{D}$. Then, we have:* $\mathbb{E}[r] := 2 \cdot (N - T_\alpha) \cdot \frac{M}{N}$, *for any error distribution $\mathcal{D}$.*

Note that in HashMark$_1$ where $N < M$, the number of tries needed for the adversary is inversely proportional to $N$, making HashMark$_1$ more robust to noise addition attacks. Meanwhile, in HashMark$_2$, since $M = N$, the number of tries needed is much smaller. Consequently, one can envision HashMark$_2$ where only a specific subset of cells (chosen at random) is embedded with the bit. While this makes it more resilient to modification attacks, the problem of efficiently identifying this subset of cells becomes paramount.

**Other Attacks.** We also consider two additional attack vectors:

- **Data augmentation:** Adding rows lowers the $z$-score. Since the secret is unknown, about half of the added rows will map to 0 on average. For instance, doubling $m$ valid rows reduces the $z$-score by a factor of $\sqrt{2}$ in expectation.

- **Feature selection:** The $z$-score threshold depends on the number of columns (Section 5.1.1). Removing columns thus requires raising the detection threshold.

HashMark **and Applications.** Watermarking tabular data enables verifiable integrity in organizational settings where datasets are routinely shared. With HashMark$_2$, two guarantees hold when a watermark is detected in a dataset $D$: (1) Theorem 3 bounds the expected number of undetectable cell modifications, and (2) if an attacker injects $\gamma m$ rows into an $m$-row dataset, the $z$-score degrades predictably, scaling as $\sqrt{1 + \gamma}$. These properties define a measurable trust boundary, supporting provenance tracking while tolerating benign changes. By formalizing this robustness–utility tradeoff, our work advances watermarking for practical data governance.

## 5 EXPERIMENTAL RESULTS

In this section, we focus on experimentation for embedding watermarks in numerical data, specifically floating-point values. Our experiments were performed on an Apple MacBook M1 Pro with 16GB of memory running Sonoma 14.3. We used Python 3.11. We instantiated the hash function using SHA-256 from the hashlib module. We select a random seed for evaluating the hash function. We implemented Generate by adding $10^{-c}$ to the value until it hashes to 0. Our choice of $c$ is specified for each context separately. Due to space constraints, we will focus on HashMark$_2$ in this section. We defer the experiments pertaining to HashMark$_1$ to the appendix in Section D.1

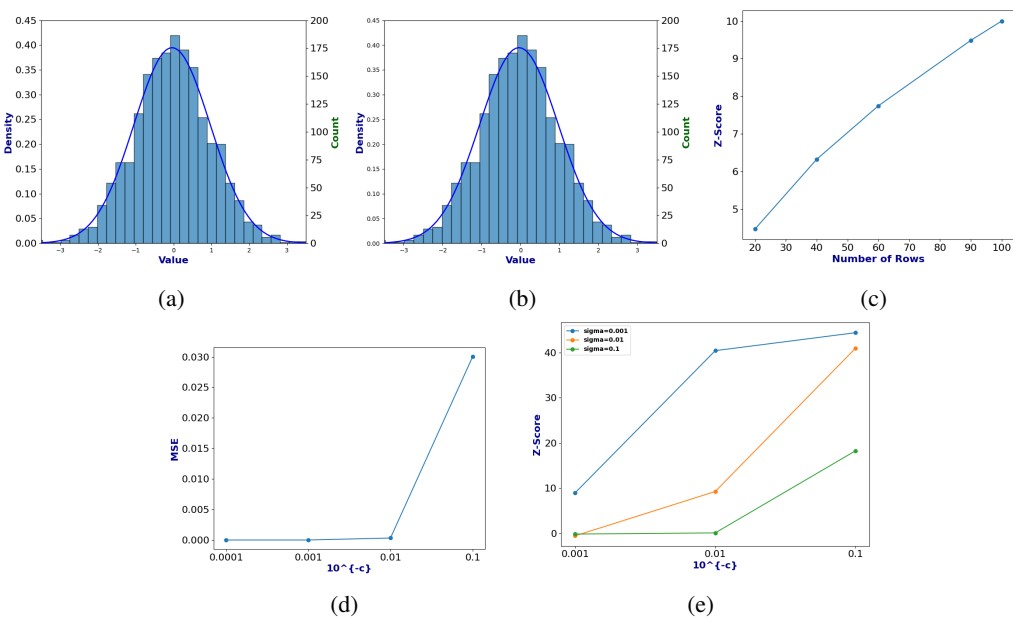

Figure 2: Plot of various experiments on Gaussian dataset. Figures 2a and 2b show the distribution of the data, before and after watermarking. Value refers to the actual value in the dataset. Figure 2c shows the variation of the z-score with the number of rows sampled. Figure 2d plots the variation of the mean-squared error (MSE) for different choices of $c$. Figure 2e plots the change in z-score when compared with the choice of $c$ for various Gaussian noises.

### 5.1 EVALUATION OF HashMark$_2$

In this section, we evaluate the performance of HashMark$_2$ along the following dimensions with additional details on the experimental setup found in Section D.2.1:

**Performance (vs the work of Ngo et al. (2024)) on Gaussian Datasets.** Following Ngo et al. (2024), we evaluate HashMark$_2$ on Gaussian data (1 column, 2000 rows). With $c = 10$, HashMark$_2$ achieves comparable robustness and fidelity while being significantly simpler, showing that complex watermarking is unnecessary.

- **Fidelity:** KDE plots (Figs.2a–2b) show near-identical distributions pre- and post-watermarking. Figure 2d confirms that smaller $c$ values (larger perturbations) increase MSE, as expected.

- **Robustness:** Figure 2c shows z-scores increase with more rows, strengthening detection. Under added Gaussian noise (Fig.2e), smaller $c$ values lower z-scores, indicating higher sensitivity. Importantly, our z-scores consistently exceed those of Ngo et al. (Fig.6). Extended results (Figs.8a, 8b) in the appendix confirm these trends.

  For completeness, Figure7 reproduces Ngo et al.'s plots, while Figures 8a–8b provide additional HashMark$_2$ results, all consistent with the conclusions above.

**Utility for Real-Life Datasets.** Following prior work (He et al., 2024; Ngo et al., 2024), we evaluate HashMark$_2$ on four real-world datasets (Section B), training CTGAN (Xu et al., 2019), Gaussian Copula (Masarotto & Varin, 2012), and TVAE (Xu et al., 2019) via the Synthetic Data Vault (Patki et al., 2016). Table 2b shows that watermarking minimally affects accuracy, even for multi-class tasks. We also study constrained sampling, where rows are retained only if at least a fraction $t$ of columns hash to 0. Tables 4–6 show that larger $t$ increases generation time but preserves accuracy, with $z$-scores rising as expected; similar trends hold for regression ($R^2$).

**Fidelity for Alphanumeric Synthetic Data.** We assess HashMark$_2$ on alphanumeric attributes by computing the Jensen–Shannon divergence (JSD) between watermarked synthetic data (all values hash to 0) and real datasets, using SciPy's implementation Virtanen et al. (2020) over 30 trials:

- **ASINs** (10-character alphanumeric): $0.1090 \pm 0.0016$ JSD vs. Amazon Product Dataset PromptCloud (2020)
- **Git commit hashes** (40-character hex): $0.002176 \pm 0.0003$ JSD vs. GitHub Commit Messages Dave (2023)

The consistently low JSD values show that HashMark$_2$ preserves the underlying distributions, even for alphanumeric data.

**Simpler Classifiers and Datasets.** To assess HashMark$_2$ in a simpler setting, we evaluate it on single-attribute, two-class datasets using linear regression, logistic regression, and decision trees. Results in Table 2a show that while the perturbation parameter ($10^{-c}$) governs the deviation from the original values, even small $c$ values lead to only negligible changes in model performance.

Table 2: Performance of HashMark$_2$ with Generate instantiated by incrementing with $10^{-c}$.

(a) Performance with Simple Regression Models. W/M = Watermarked dataset. For Logistic/Decision Tree, we report accuracy; for Linear Regression, we report $R^2$ values.

|  | $c=2$ | $c=4$ | $c=6$ |
|---|---|---|---|
| **Logistic Reg. (Orig.)** | 99.98% | 99.98% | 99.98% |
| **Logistic Reg. (W/M)** | 99.64% | 99.98% | 99.98% |
| **Linear Reg. (Orig. $R^2$)** | 1.000000 | 1.000000 | 1.000000 |
| **Linear Reg. (W/M $R^2$)** | 0.999899 | 1.000000 | 1.000000 |
| **Decision Tree (Orig.)** | 100% | 100% | 100% |
| **Decision Tree (W/M)** | 100% | 99.995% | 99.961% |

(b) Accuracy comparison of different classifiers and synthesizers across four datasets on synthetic and watermarked synthetic data. Standard deviations are included for each record. W/M = Watermarked synthetic dataset, while Non-W/M refers to an unwatermarked but synthetic dataset. Here, $c = 6$.

| Dataset | Classifier | Synth. | Non-W/M (%) | W/M (%) |
|---|---|---|---|---|
| Wilt | XGB | CTGAN | $83.63 \pm 4.63$ | $83.31 \pm 5.01$ |
|  |  | Copula | $94.38 \pm 0.53$ | $94.40 \pm 0.52$ |
|  |  | TVAE | $94.87 \pm 0.37$ | $94.89 \pm 0.39$ |
|  | RF | CTGAN | $84.45 \pm 5.74$ | $84.30 \pm 5.70$ |
|  |  | Copula | $94.39 \pm 0.52$ | $94.40 \pm 0.52$ |
|  |  | TVAE | $94.34 \pm 0.37$ | $94.34 \pm 0.38$ |
| Housing | XGB | CTGAN | $49.26 \pm 2.38$ | $49.11 \pm 2.68$ |
|  |  | Copula | $55.15 \pm 5.12$ | $55.66 \pm 4.77$ |
|  |  | TVAE | $61.55 \pm 2.39$ | $61.13 \pm 2.46$ |
|  | RF | CTGAN | $48.31 \pm 1.90$ | $48.14 \pm 2.00$ |
|  |  | Copula | $52.97 \pm 5.83$ | $53.04 \pm 5.93$ |
|  |  | TVAE | $62.30 \pm 1.92$ | $62.40 \pm 1.77$ |
| HOG | XGB | CTGAN | $77.65 \pm 2.07$ | $77.62 \pm 2.08$ |
|  |  | TVAE | $89.77 \pm 1.59$ | $89.34 \pm 1.76$ |
|  | RF | CTGAN | $74.40 \pm 4.41$ | $74.39 \pm 4.48$ |
|  |  | TVAE | $91.20 \pm 2.16$ | $91.28 \pm 2.16$ |
| Shoppers | XGB | CTGAN | $86.43 \pm 0.79$ | $85.28 \pm 1.95$ |
|  |  | Copula | $86.01 \pm 1.38$ | $86.56 \pm 1.41$ |
|  |  | TVAE | $87.94 \pm 0.61$ | $87.85 \pm 0.54$ |
|  | RF | CTGAN | $87.77 \pm 0.82$ | $86.00 \pm 2.74$ |
|  |  | Copula | $86.05 \pm 1.40$ | $85.78 \pm 1.38$ |
|  |  | TVAE | $88.71 \pm 1.00$ | $88.10 \pm 1.23$ |

# 6 CONCLUSION

We present HashMark, a hash-based framework for watermarking tabular datasets, enhancing data integrity, provenance, and accountability in machine learning pipelines. HashMark supports both numerical and categorical features, improving upon prior approaches (He et al., 2024; Ngo et al., 2024; Zheng et al., 2024) while maintaining downstream utility. Our method naturally extends to synthetic data, enabling the verifiable and responsible use of generative models in applications such as stress testing, privacy-preserving data sharing, and benchmark creation. By providing rigorous fidelity guarantees and addressing challenges in correlation preservation, HashMark contributes to ongoing efforts in secure data management, trustworthy machine learning, and the development of robust datasets and benchmarks for future research.

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

## A    RELATED WORK

**Watermarking Tabular Data.**    Watermarking tabular data has been extensively studied. Agrawal & Kiernan (2002) pioneered a scheme embedding watermarks in the least significant bit of specific cells using hash values based on primary and private keys. Subsequent works by Xiao et al. (2007) and Hamadou et al. (2011) improved this by embedding multiple bits. Another approach embeds watermarks in statistical properties. Sion et al. (2003) introduced a method that partitions dataset rows and modifies subset statistics, later refined by Shehab et al. Shehab et al. (2008) to resist insertion and deletion attacks using optimized partitioning and hash-based embedding. Their approach, however, relies on assumptions about data distribution and primary keys.

Inspired by watermarking techniques in large language models Aaronson (2023); Kamaruddin et al. (2018); Kirchenbauer et al. (2023), He et al. (2024), Ngo et al. (2024), and Zheng et al. (2024) proposed watermarking schemes for generative tabular data using red-green interval partitioning.

He et al. (2024) introduced a data binning approach, ensuring values lie near green intervals and using statistical hypothesis testing for detection. However, assuming continuous distributions makes it vulnerable to feature selection and truncation attacks. Ngo et al. (2024) paired columns into key-value sets, deriving a seed from the key column to generate bins for the value column. Entries falling in red bins were resampled from green bins. While novel, this method suffers from two key weaknesses: (i) detection requires prior knowledge of the column pairing or an exhaustive search across all pairs, and (ii) relying on key column-derived seeds introduces low entropy, weakening the pseudorandomness of bin assignments and potentially compromising security. It is important to note that even with knowledge of column pairing, any deletion of rows will trigger an error when calculating the key column-derived seed, which is not explored or discussed in the paper. Zheng et al. (2024) took a similar approach, embedding watermarks as additive noise within predefined bins. They assumed noise follows a bounded range $[-p, p]$, partitioned into red and green bins, with watermarking achieved by sampling noise only from green bins. Despite robustness claims and categorical feature support, their method has several limitations. First, detection requires access to the original dataset, making watermark verification infeasible in practical scenarios where datasets are modified or shuffled. Second, row-matching under permutation increases detection complexity. Finally, their claimed support for categorical data is unclear and lacks empirical validation - (a) Their protocol description focuses only on categorical data, i.e., those with a fixed range (e.g., education level, employee designation, marital status, etc.). They suggest encoding it first as integers and then applying their embedding techniques. However, this method is flawed because these differences often result in floating-point values, distorting the expected integer-based distribution. Restricting differences to integers could also leave gaps in the data (by omitting particular values from the range), harming its utility. Instead, we argue against watermarking such columns altogether, and (b) it does not address unrestricted categorical data (e.g., alphanumeric ASINs) or provide experiments for such cases. The above is summarized in Table 1.

**Watermarking for LLMs.**    Many watermarking schemes for LLMs take advantage of the sampling algorithm that generates each token of an LLM output. Christ et al. (2024) observed that these LLM output tokens correlate with the randomness used in the token sampling algorithm. This correlation is efficiently communicable for many LLM outputs by replacing this randomness with cryptographic pseudorandomness. Subsequent works Fairoze et al. (2025); Christ & Gunn (2024) have built upon this idea by incorporating error correction and public identifiability into these watermarks. However, robustness remains a persistent issue for this line of work, and a recent impossibility result Zhang et al. (2024a) demonstrated that an adversary that can efficiently perturb or resample the output can always remove a watermark. Another line of work, which has been the source of inspiration for more recent watermarking schemes for tabular data, include Aaronson (2023); Kamaruddin et al. (2018); Kirchenbauer et al. (2023). Kirchenbauer et al. (2023) introduced the red-green list paradigm, forming the basis of several works He et al. (2024); Zheng et al. (2024); Ngo et al. (2024). More recently, Giboulot & Furon (2024) improved on the works employing the red-green list paradigm.

## B    DATASET DETAILS

**Wilt.**    Wilt (Johnson, 2013) is the public dataset from the UCI Machine Learning Repository from a remote sensing study on detecting diseased trees in satellite imagery. It comprises 4,839

image segments with spectral and texture features from Quickbird multispectral and panchromatic bands. The dataset includes six numerical and categorical attributes and a binary classification task: identifying trees as wilted or healthy. We generate synthetic datasets. There are 4839 records with 6 features (including the target) and 2 classes. This dataset is licensed under a Creative Commons Attribution 4.0 International (CC BY 4.0) license.

**California Housing Prices.** The California Housing Prices dataset (Kelley Pace & Barry, 1997; Géron, 2019), sourced from the 1990 U.S. Census, contains 20,640 records with 10 socio-economic and geographical attributes influencing housing prices. It has a multi-target label indicating proximity to the ocean, making it a multi-class classification problem. It has 5 classes. This dataset is licensed under Apache License Version 2.0.

**HOG.** The HOG feature dataset (Alpaydin & Alimoglu, 1996) is generated with the histogram of oriented gradients (HOG) features extracted from the digits dataset, combined with their categories. There are 16 features, 10992 records, and 10 classes. This dataset is licensed under a Creative Commons Attribution 4.0 International (CC BY 4.0) license.

**Shoppers Dataset.** The shoppers dataset (Sakar et al., 2018) aimed to capture the shoppers purchasing intent. There are 12,330 records with 18 attributes with two classes. The dataset is licensed under a Creative Commons Attribution 4.0 International (CC BY 4.0) license.

**King Dataset.** The King dataset (harlfoxem, 2016) aimed to capture the prices of house sales for King County, which includes Seattle. It includes homes sold between May 2014 and May 2015. It is useful for regression-based price prediction. There are 21,613 rows records with 12 columns[2] with the price column being the regression data target. The dataset is licensed under a CC0 license.

**Amazon ASINs.** We used the Amazon Product Details Dataset **?**. For our experiments, we parsed the dataset only to extract the unique identifiers for Amazon products, generating 30,000 actual ASINs. This dataset is licensed under CC0.

**Gitcommit Hashes.** We used the Gitcommit Messages dataset (Dave, 2023). It contains 4.3 million records, from which we only extracted the hashes for the gitcommit messages. The dataset is licensed under the Open Data Commons Attribution License (ODC-By) v1.0.

## C  HashMark$_1$: Embedding Pseudorandom Bits

We begin by describing our first approach to watermarking. This approach ensures high fidelity and detectability but suffers from issues when it comes to robustness. The embedding algorithm is formally defined in Algorithm 2. We start with an original dataset $\mathbf{X}$ of dimension $m \times n$. The idea is to sample $\ell$ pseudorandom bits. Let us call it $bit_1, \ldots, bit_\ell$. Additionally, we also sample $\ell$ cells defined by $(row_i, col_i)$ in $\mathbf{X}$. By modifying the cell content suitably, we ensure that $\mathcal{H}(\mathbf{X}[row_i, col_i]) = bit_i$.

### C.1  Detecting HashMark$_1$

To detect, the algorithm needs:

- Knowledge of $X_1$ to retrieve the original binary string of $bit_1, \ldots, bit_\ell$.
- Knowledge of $X_2$ to first identify the target cells $(row_i, col_i)$, and then using $\mathcal{H}$ to retrieve $bit'_1, \ldots, bit'_\ell$.
- The watermark detection is successful iff $(bit_1, \ldots, bit_\ell) = (bit'_1, \ldots, bit'_\ell)$

However, this scheme is low-robust because the detection algorithm critically relies on extracting the cell where the watermark was embedded. This would be meaningless if the first row (or the first column) were removed. The benefit of this approach is that only $\ell$ of the spots are touched, which is

---

[2]We used only 12 columns for our experiments. These are "floors", "waterfront","lat" ,"bedrooms" ,"sqft_basement" ,"view" ,"bathrooms","sqft_living15","sqft_above","grade","sqft_living","price".

---

**Algorithm 2** HashMark$_1$ Embedding Algorithm

---

**Input:** Original Dataset $\mathbf{X}$ of dimension $m \times n$
Probability Distributions $\rho_1, \ldots, \rho_n$.
PRG $G_1 : \mathcal{X}_1 \to \{0,1\}^{\ell}$
PRG $G_2 : \mathcal{X}_2 \to [m]^{\ell} \times [n]^{\ell}$
$X_1 \overset{\$}{\leftarrow} \mathcal{X}_1, X_2 \overset{\$}{\leftarrow} \mathcal{X}_2$
$\{bit_i\}_{i=1}^{\ell} \overset{\$}{\leftarrow} G_1(X_1)$
$\{(row_i, col_i)\}_{i=1}^{\ell} \leftarrow G_2(X_2)$
$seed \overset{\$}{\leftarrow} \mathcal{S}$ //$\mathcal{S}$ is the seed space of $\mathcal{H}$

---

**for** $i = 1$ **to** $\ell$ **do**
   **while** $\mathcal{H}(seed, \mathbf{X}[row_i, col_i]) \neq bit_i$ **do**
      $new\_value \overset{\$}{\leftarrow} \mathsf{Generate}(\rho_i, \mathbf{X}[row_i, col_i])$
      $\mathbf{X}[row_i, col_i] \leftarrow new\_value$
   **end while**
**end for**

---

a tunable parameter. This ensures very high fidelity and utility. The detectability is also reducible to the hardness of the underlying cryptographic primitives (and does not rely on a statistical measure).

## D  EXPERIMENTS FOR HashMark

### D.1  EVALUATION OF HashMark$_1$

We begin by benchmarking the performance of HashMark$_1$ along the following axes:

- Varying $\ell$, we wish to study the running time of the watermarking process. We break down the running time of watermarking as (a) the cost of identifying locations to embed the watermark and (b) the time taken to run Generate to embed the desired bits.
- The utility of the watermarked dataset vs. the original dataset for downstream machine learning tasks.
- The role of $\ell$ in accuracy, i.e., how does the accuracy change when more bits are embedded?

**Performance of Embedding Process.**    In Figure 3, we plot the time, in seconds, against the number of bits being embedded. We split the cost as follows: to generate locations for embedding (dubbed pair generation time) and then modify the cell content until it hashes to the desired bit. Recall that the pair generation time requires using a seed to produce $\ell$ cell positions, which only contain floating point values. We then use the same seed to generate $\ell$ bits additionally. As one can observe, the embedding time is much smaller than the pair generation time, and it takes less than 10 milliseconds to embed as many as 1000 bits.

**Dataset.**    We study the above for a specific dataset - the adult census income dataset from (Byrd et al., 2022; Jayaraman et al., 2018) to predict if an individual earns over \$50,000 per year. The preprocessed dataset has 105 features and 45,222 records with a 25% positive class (i.e., 25% of the records have class 1 while the rest are in class 0) We randomly split into training and testing datasets. We observed that the dataset consisted of integers or floating-point values with at least eight decimal places. This leads us to choose $c = 6$ and embed it only in the floating-point values.

**Downstream Utility.**    We embed $\ell = 384$ bits [3] They are:

- Logistic Regression Classifier with maximum iterations as 1000

---

[3]Choice of $\ell$ is set to be 384 because it is the number of bits in a standard hash-based watermarking scheme, albeit for messaging applications (i.e., signatures) known as BLS Signature Boneh et al. (2004). Note that this corresponds to less than 1% of the number of cells in the dataset.

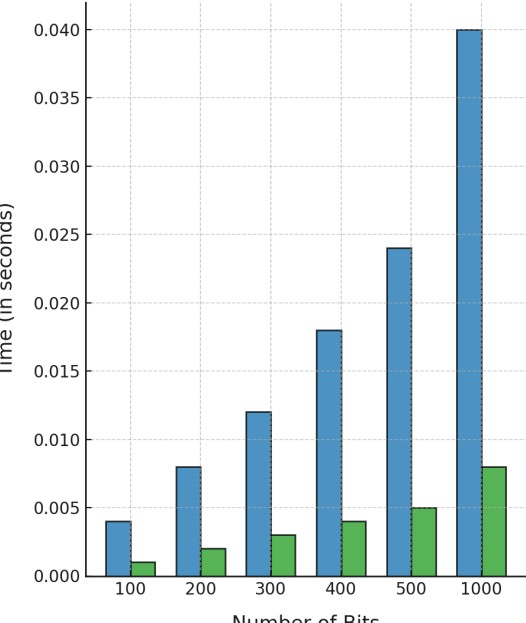

Figure 3: Embedding Time as a function of $\ell$ for $\mathsf{HashMark}_1$. Here, the blue column refers to the cost of generating valid cells to embed in the dataset, while the green column is the cost of modifying the content to make it hash to the desired bit.

Table 3: Classification accuracy (%) with and without watermarking. In addition to this, we add the standard deviation of each record.

| Model | Logistic Regression | Random Forest | MLP Classifier |
|---|---|---|---|
| Original | $84.021 \pm 0.3$ | $85.186 \pm 0.27$ | $83.504 \pm 0.44$ |
| Watermarked | $84.021 \pm 0.3$ | $85.188 \pm 0.28$ | $83.508 \pm 0.446$ |

- Random Forest Classifier with 100 estimators
- MLP Classifier with hidden layer sizes 100, 50; maximum iterations=1000, and learning rate 0.0001

We plotted the difference in accuracy when run on the original versus the watermarked dataset in Figures 4 and 5 for each of the 1000 runs. Meanwhile, in Table 3, we present the average accuracy of the 1000 runs. Identical behavior was observed in the Logistic Regression classifier with less than $0.005\%$ difference observed in the accuracy of the other two classifiers. This shows that $\mathsf{HashMark}_1$'s embedding has a negligible impact on the accuracy of the classifier. For completeness, we also plot the difference in accuracy between the original and watermarked dataset in Figures 4 and 5, in each of the 1000 runs. As can be observed, the most significant difference in accuracy is less than $0.005\%$.

Finally, in Figure 5b, we plot the impact of increasing $\ell$ on the accuracy of the logistic regression classifier. As expected, larger $\ell$ does cause an impact in accuracy, though the degradation is minimal.

## D.2 EVALUATION OF $\mathsf{HashMark}_2$

### D.2.1 EXPERIMENTAL SETUP

We now present additional details about the experimental setup for the various experiments described in Section 5.1. In all cases, default hyperparameters were selected unless otherwise specified.

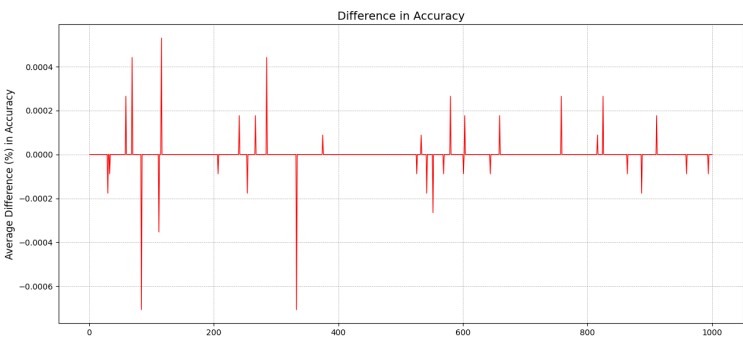

(a) Plot of the difference in accuracy between the original and the watermarked dataset in each of the 1000 iterations for the Logistic Regression Classifier

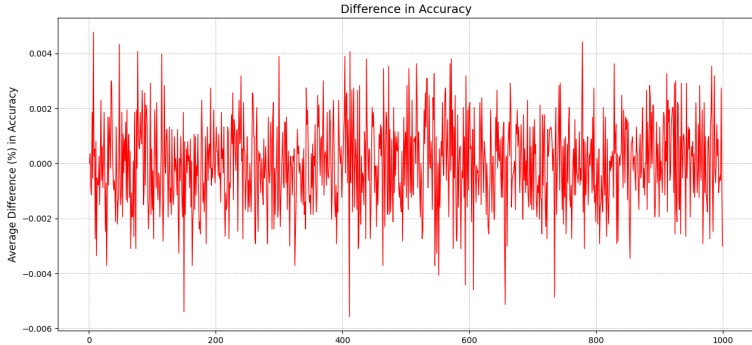

(b) Plot of the difference in accuracy between the original and the watermarked dataset in each of the 1000 iterations for the Random Forest Classifier

Figure 4: Experiments pertaining to $\mathsf{HashMark}_1$ for the Adult Census Dataset (Part 1).

- Gaussian Datasets: We follow the same experimental setup as described by Ngo et al. (2024), with the only exception that we rely on 1 column (as opposed to 2 in their work, which is necessitated by their protocol description). The Generate function is defined by adding $10^{-c}$ until it hashes to 0. We conduct various studies, as documented in Figure 2, with additional experiments described below in Figure 6 and Figure 8.

- Utility for Real-Life Datasets: We evaluate our approach on four classification datasets and one regression dataset, following prior work. Each dataset is split 75/25 into training and test sets. Using the Synthetic Data Vault (SDV), we train three generative models with default hyperparameters: CTGAN (GAN-based), Gaussian Copula (copula-based), and TVAE (VAE-based). Once synthetic data is generated, we train machine learning models on it—two classifiers for the classification datasets and three regressors for the regression dataset—and measure performance using classification accuracy (for classification) or $R^2$ (for regression). Performance was measured with respect to the *original* test data.

  We then embed watermarks into the generated synthetic datasets. For classification datasets, we evaluate two watermarking strategies:

  1. Perturbation-based Generate: adding a small perturbation of $10^{-6}$.
  2. Constrained sampling-based Generate: rows are drawn i.i.d. from the learned distribution $\rho$ and retained only if at least a fraction $t$ of the $n$ columns hash to 0; otherwise, the row is resampled until the dataset reaches the target size.

  For the regression dataset, we only apply the constrained sampling approach. After watermarking, we retrain the downstream ML models and measure accuracy/$R^2$, compute $z$-scores to assess watermark detectability, and record the average watermarking time for

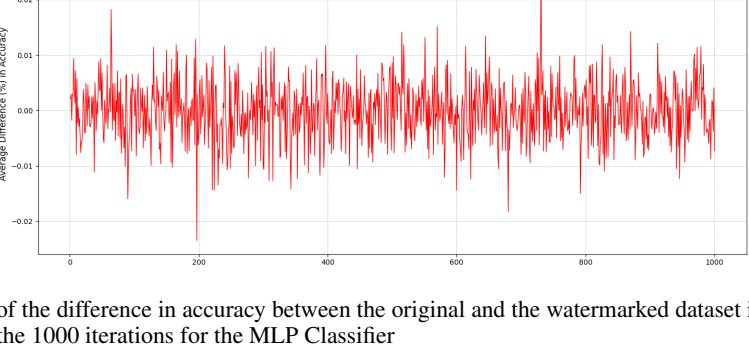

(a) Plot of the difference in accuracy between the original and the watermarked dataset in each of the 1000 iterations for the MLP Classifier

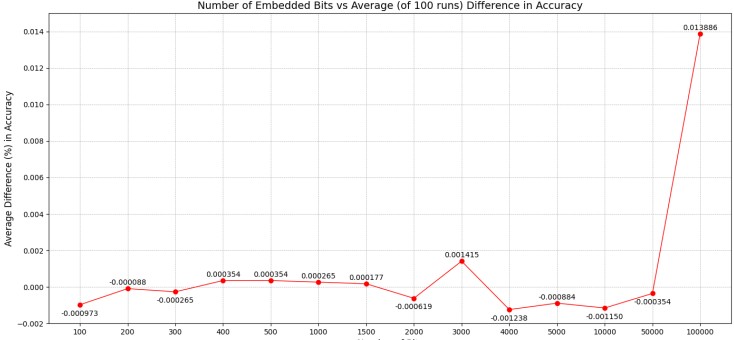

(b) Average Difference in the accuracy of the logistic regression classifier as the number of bits embedded ($\ell$) increases.

Figure 5: Experiments pertaining to $\mathsf{HashMark}_1$ for the Adult Census Dataset (Part 2).

the rejection-sampling method. Performance was measured with respect to the *original* test data.

- Fidelity of Alphanumeric Synthetic Data: We generate random 10-character of alphanumeric for ASINs and 40-characters of hexadecimal values such that they all hash to 0. We then take an average of the 30 trials, measuring the Jensen-Shannon Divergence.

We also present additional experiments studying the variation of MSE with respect to the choice of $c$ for further values of $c$. Similarly, we also show how the Z-score varies for larger sampled rows. This is done in Figure 8.

In Figure 7, we reproduce Figure 2 from Ngo et al. Ngo et al. (2024). This shows that the performance of $\mathsf{HashMark}_2$, as seen in Figure 2, matches (or surpasses) similar experiments from Ngo et al. This is especially important, considering that $\mathsf{HashMark}_2$ is conceptually simpler, offers support for categorical data, and is more secure. Recall that $\mathsf{HashMark}_2$ uses a truly random value as a seed, while Ngo et al. opt for a heuristic approach to obtain a seed via a pairing algorithm, which are often poor sources of entropy.

# E  DEFERRED PROOFS

*Proof of Theorem 1.* For each element $x_i$ in $\mathbf{X}$, let $x_i'$ be the corresponding element in $\mathbf{X}_w$. As defined above:
$$x_i' = x_i + k_i \cdot 10^{-c},$$
where $k_i = \min\{k \geq 0 \mid \mathcal{H}(x_i + k \cdot 10^{-c}) = 0$. In other words, $|x_i - x_i'| = k_i \cdot 10^{-c}$.

Table 4: Effect of constraint threshold $t$ on synthetic data quality across two datasets. We report the average $z$-scores, sampling time (in seconds), and classification accuracy (in %) using different classifiers and synthesizers. This is with respect to HashMark$_2$. Accuracy is shown for both the non-W/M and W/M settings.

| Dataset | $t$ | $z$-score | Sampling Time (s) | Classifier | Synthesizer | Non-W/M (%) | W/M (%) |
|---|---|---|---|---|---|---|---|
| Wilt 3629 samples 5 columns | 1/4 | 1.74 ± 0.22 | 64.08 ± 6.68 | XGB | TVAE | 95.24 ± 0.57 | 95.07 ± 0.84 |
| | | | | | GC | 94.33 ± 0.31 | 94.53 ± 0.24 |
| | | | | | CTGAN | 83.65 ± 3.24 | 81.79 ± 6.76 |
| | | | | RF | TVAE | 94.78 ± 0.30 | 94.86 ± 0.42 |
| | | | | | GC | 94.43 ± 0.31 | 94.45 ± 0.30 |
| | | | | | CTGAN | 84.31 ± 1.93 | 84.76 ± 1.54 |
| | 1/3 | 1.92 ± 0.24 | 65.45 ± 4.90 | XGB | TVAE | 94.86 ± 0.44 | 95.19 ± 0.43 |
| | | | | | GC | 94.33 ± 0.31 | 94.53 ± 0.24 |
| | | | | | CTGAN | 84.07 ± 4.94 | 85.62 ± 6.19 |
| | | | | RF | TVAE | 94.84 ± 0.45 | 94.76 ± 0.66 |
| | | | | | GC | 94.43 ± 0.31 | 94.45 ± 0.30 |
| | | | | | CTGAN | 86.08 ± 6.52 | 86.60 ± 6.82 |
| | 1/2 | 9.43 ± 0.44 | 65.05 ± 1.26 | XGB | TVAE | 95.02 ± 0.44 | 95.31 ± 0.48 |
| | | | | | GC | 94.33 ± 0.31 | 94.43 ± 0.33 |
| | | | | | CTGAN | 82.55 ± 7.06 | 84.23 ± 6.56 |
| | | | | RF | TVAE | 94.73 ± 0.43 | 94.68 ± 0.45 |
| | | | | | GC | 94.43 ± 0.31 | 94.43 ± 0.30 |
| | | | | | CTGAN | 80.46 ± 7.11 | 80.50 ± 6.03 |
| | 2/3 | 22.73 ± 0.30 | 108.95 ± 4.59 | XGB | TVAE | 95.22 ± 0.32 | 95.17 ± 0.65 |
| | | | | | GC | 94.33 ± 0.31 | 94.26 ± 0.46 |
| | | | | | CTGAN | 78.83 ± 9.36 | 78.86 ± 9.80 |
| | | | | RF | TVAE | 94.83 ± 0.66 | 94.83 ± 0.25 |
| | | | | | GC | 94.43 ± 0.31 | 94.41 ± 0.34 |
| | | | | | CTGAN | 82.12 ± 5.57 | 84.21 ± 5.18 |
| | 3/4 | 23.20 ± 0.16 | 116.91 ± 9.98 | XGB | TVAE | 95.21 ± 0.32 | 95.32 ± 0.53 |
| | | | | | GC | 94.33 ± 0.31 | 94.26 ± 0.46 |
| | | | | | CTGAN | 78.38 ± 6.28 | 77.47 ± 6.41 |
| | | | | RF | TVAE | 94.93 ± 0.41 | 94.84 ± 0.29 |
| | | | | | GC | 94.43 ± 0.31 | 94.41 ± 0.34 |
| | | | | | CTGAN | 79.87 ± 6.53 | 80.74 ± 5.64 |
| Housing 15480 samples 9 columns | 1/4 | 2.84 ± 0.72 | 449.17 ± 40.27 | XGB | TVAE | 63.35 ± 0.76 | 63.43 ± 0.79 |
| | | | | | GC | 52.82 ± 3.99 | 52.27 ± 3.24 |
| | | | | | CTGAN | 47.07 ± 2.58 | 46.59 ± 2.15 |
| | | | | RF | TVAE | 62.79 ± 0.59 | 62.93 ± 0.43 |
| | | | | | GC | 53.60 ± 2.02 | 53.71 ± 3.27 |
| | | | | | CTGAN | 45.72 ± 2.02 | 46.50 ± 2.31 |
| | 1/3 | 2.63 ± 0.64 | 415.68 ± 7.37 | XGB | TVAE | 62.75 ± 1.54 | 62.86 ± 1.42 |
| | | | | | GC | 52.82 ± 3.99 | 52.27 ± 3.24 |
| | | | | | CTGAN | 46.48 ± 1.73 | 46.63 ± 2.89 |
| | | | | RF | TVAE | 61.91 ± 2.63 | 61.93 ± 2.29 |
| | | | | | GC | 53.60 ± 2.02 | 53.71 ± 3.27 |
| | | | | | CTGAN | 48.99 ± 1.39 | 48.69 ± 1.20 |
| | 1/2 | 18.27 ± 0.20 | 552.12 ± 12.20 | XGB | TVAE | 60.95 ± 3.12 | 61.02 ± 3.05 |
| | | | | | GC | 52.82 ± 3.99 | 52.76 ± 2.63 |
| | | | | | CTGAN | 47.70 ± 1.95 | 48.75 ± 3.12 |
| | | | | RF | TVAE | 63.38 ± 0.16 | 63.30 ± 0.45 |
| | | | | | GC | 53.60 ± 2.02 | 53.45 ± 2.90 |
| | | | | | CTGAN | 49.81 ± 2.78 | 47.59 ± 3.02 |
| | 2/3 | 34.43 ± 0.29 | 848.09 ± 17.84 | XGB | TVAE | 61.75 ± 2.03 | 61.88 ± 1.73 |
| | | | | | GC | 53.60 ± 4.82 | 52.91 ± 2.85 |
| | | | | | CTGAN | 47.77 ± 2.52 | 46.29 ± 3.79 |
| | | | | RF | TVAE | 62.24 ± 1.30 | 62.24 ± 1.55 |
| | | | | | GC | 54.06 ± 2.88 | 53.74 ± 3.21 |
| | | | | | CTGAN | 48.81 ± 2.08 | 48.13 ± 2.04 |
| | 3/4 | 53.74 ± 0.29 | 1632.13 ± 79.29 | XGB | TVAE | 62.13 ± 1.85 | 62.83 ± 1.97 |
| | | | | | GC | 52.82 ± 3.99 | 53.91 ± 3.73 |
| | | | | | CTGAN | 46.84 ± 3.37 | 48.00 ± 2.20 |
| | | | | RF | TVAE | 60.86 ± 2.19 | 60.87 ± 2.20 |
| | | | | | GC | 53.60 ± 2.02 | 53.75 ± 2.66 |
| | | | | | CTGAN | 49.89 ± 2.68 | 48.56 ± 1.73 |

Table 5: Effect of constraint threshold $t$ on synthetic data quality across two datasets. We report the average $z$-scores, sampling time (in seconds), and classification accuracy (in %) using different classifiers and synthesizers. This is with respect to HashMark$_2$. Accuracy is shown for both the non-W/M and W/M settings.

| Dataset | $t$ | $z$-score | Sampling Time (s) | Classifier | Synthesizer | Non-W/M (%) | W/M (%) |
|---|---|---|---|---|---|---|---|
| HOG 8244 samples 18 columns | 1/4 | -5.77 ± 0.78 | 373.24 ± 105.90 | XGB | TVAE | 88.52 ± 4.74 | 87.53 ± 4.89 |
| | | | | | CTGAN | 73.74 ± 3.15 | 72.92 ± 3.75 |
| | | | | RF | TVAE | 92.48 ± 1.33 | 93.03 ± 1.22 |
| | | | | | CTGAN | 74.56 ± 3.28 | 74.24 ± 2.90 |
| | 1/3 | -4.89 ± 1.15 | 511.61 ± 8.76 | XGB | TVAE | 88.84 ± 1.90 | 90.64 ± 1.19 |
| | | | | | CTGAN | 70.44 ± 6.26 | 70.87 ± 5.59 |
| | | | | RF | TVAE | 91.36 ± 0.94 | 91.92 ± 1.00 |
| | | | | | CTGAN | 73.29 ± 3.81 | 73.25 ± 4.12 |
| | 1/2 | 7.46 ± 0.18 | 797.56 ± 12.58 | XGB | TVAE | 91.43 ± 1.27 | 91.49 ± 0.85 |
| | | | | | CTGAN | 75.44 ± 3.19 | 75.82 ± 3.40 |
| | | | | RF | TVAE | 92.43 ± 1.01 | 91.86 ± 0.89 |
| | | | | | CTGAN | 74.32 ± 3.27 | 74.00 ± 3.09 |
| | 2/3 | 31.40 ± 0.21 | 9868.32 ± 8790.14 | XGB | TVAE | 88.80 ± 2.53 | 87.78 ± 2.71 |
| | | | | | CTGAN | 72.71 ± 2.93 | 73.34 ± 3.31 |
| | | | | RF | TVAE | 91.78 ± 1.63 | 91.47 ± 2.50 |
| | | | | | CTGAN | 72.31 ± 3.75 | 72.71 ± 4.08 |
| | 3/4 | 40.77 ± 0.16 | 35088.75 ± 30542.58 | XGB | TVAE | 90.83 ± 1.00 | 90.74 ± 1.09 |
| | | | | | CTGAN | 75.26 ± 4.04 | 74.95 ± 4.00 |
| | | | | RF | TVAE | 88.92 ± 3.03 | 88.08 ± 3.70 |
| | | | | | CTGAN | 70.71 ± 5.23 | 70.20 ± 5.15 |
| Shopper 9247 samples 12 columns | 1/4 | -2.11 ± 1.38 | 438.51 ± 5.14 | XGB | TVAE | 87.78 ± 0.78 | 87.78 ± 0.76 |
| | | | | | GC | 85.51 ± 0.63 | 85.80 ± 0.76 |
| | | | | | CTGAN | 87.35 ± 0.35 | 87.06 ± 0.95 |
| | | | | RF | TVAE | 88.74 ± 0.32 | 88.74 ± 0.45 |
| | | | | | GC | 85.62 ± 0.43 | 85.99 ± 0.98 |
| | | | | | CTGAN | 87.95 ± 0.49 | 87.91 ± 0.28 |
| | 1/3 | -3.37 ± 1.26 | 639.55 ± 64.59 | XGB | TVAE | 88.13 ± 0.63 | 87.86 ± 0.86 |
| | | | | | GC | 85.51 ± 0.63 | 85.28 ± 1.17 |
| | | | | | CTGAN | 84.76 ± 1.05 | 84.94 ± 1.31 |
| | | | | RF | TVAE | 88.18 ± 0.53 | 88.06 ± 0.81 |
| | | | | | GC | 85.62 ± 0.43 | 85.70 ± 0.68 |
| | | | | | CTGAN | 88.01 ± 0.62 | 87.80 ± 0.69 |
| | 1/2 | 9.22 ± 1.27 | 939.33 ± 107.06 | XGB | TVAE | 87.27 ± 1.33 | 87.54 ± 0.94 |
| | | | | | GC | 85.51 ± 0.63 | 86.10 ± 0.94 |
| | | | | | CTGAN | 85.27 ± 1.54 | 85.59 ± 1.64 |
| | | | | RF | TVAE | 88.61 ± 0.56 | 88.28 ± 0.54 |
| | | | | | GC | 85.62 ± 0.43 | 85.65 ± 0.71 |
| | | | | | CTGAN | 87.80 ± 0.25 | 87.57 ± 0.69 |
| | 2/3 | 34.14 ± 0.28 | 3690.59 ± 252.79 | XGB | TVAE | 87.46 ± 0.69 | 88.01 ± 0.20 |
| | | | | | GC | 85.51 ± 0.63 | 85.74 ± 0.61 |
| | | | | | CTGAN | 85.89 ± 0.57 | 85.59 ± 1.70 |
| | | | | RF | TVAE | 88.48 ± 0.32 | 88.30 ± 0.64 |
| | | | | | GC | 85.62 ± 0.43 | 86.49 ± 0.56 |
| | | | | | CTGAN | 87.89 ± 0.88 | 87.82 ± 0.59 |
| | 3/4 | 43.50 ± 0.42 | 9276.41 ± 1742.76 | XGB | TVAE | 88.10 ± 0.92 | 88.39 ± 0.78 |
| | | | | | GC | 85.51 ± 0.63 | 86.06 ± 1.24 |
| | | | | | CTGAN | 86.75 ± 0.81 | 86.44 ± 0.43 |
| | | | | RF | TVAE | 88.52 ± 0.55 | 88.17 ± 0.88 |
| | | | | | GC | 85.62 ± 0.43 | 86.77 ± 0.74 |
| | | | | | CTGAN | 87.80 ± 0.58 | 87.63 ± 0.74 |

Table 6: Effect of constraint threshold $t$ on synthetic data quality for the King dataset (11 cols and 16209 samples). We report the average $z$-scores, sampling time (in seconds), and classification accuracy (as $R^2$ values) using different classifiers and synthesizers. This is with respect to HashMark$_2$. Accuracy is shown for both the non-W/M and W/M settings.

| $t$ | $z$-score | Sampling Time (s) | Classifier | Synthesizer | Non-W/M | W/M |
|---|---|---|---|---|---|---|
| 1/4 | $1.04 \pm 3.72$ | $518.84 \pm 10.49$ | Ridge | TVAE | $0.524 \pm 0.067$ | $0.518 \pm 0.074$ |
| | | | | CTGAN | $0.524 \pm 0.067$ | $0.511 \pm 0.039$ |
| | | | | GC | $0.576 \pm 0.018$ | $0.576 \pm 0.014$ |
| | | | RF | TVAE | $0.625 \pm 0.024$ | $0.614 \pm 0.011$ |
| | | | | CTGAN | $0.590 \pm 0.018$ | $0.586 \pm 0.028$ |
| | | | | GC | $0.543 \pm 0.017$ | $0.530 \pm 0.021$ |
| | | | XGB | TVAE | $0.600 \pm 0.040$ | $0.582 \pm 0.055$ |
| | | | | CTGAN | $0.575 \pm 0.019$ | $0.567 \pm 0.017$ |
| | | | | GC | $0.543 \pm 0.017$ | $0.541 \pm 0.013$ |
| 1/3 | $-0.01 \pm 3.91$ | $638.89 \pm 26.21$ | Ridge | TVAE | $0.480 \pm 0.128$ | $0.526 \pm 0.083$ |
| | | | | CTGAN | $0.507 \pm 0.083$ | $0.511 \pm 0.079$ |
| | | | | GC | $0.576 \pm 0.018$ | $0.576 \pm 0.015$ |
| | | | RF | TVAE | $0.617 \pm 0.035$ | $0.630 \pm 0.020$ |
| | | | | CTGAN | $0.558 \pm 0.047$ | $0.571 \pm 0.029$ |
| | | | | GC | $0.543 \pm 0.017$ | $0.530 \pm 0.017$ |
| | | | XGB | TVAE | $0.576 \pm 0.063$ | $0.572 \pm 0.063$ |
| | | | | CTGAN | $0.581 \pm 0.019$ | $0.573 \pm 0.017$ |
| | | | | GC | $0.543 \pm 0.017$ | $0.532 \pm 0.013$ |
| 1/2 | $16.09 \pm 1.79$ | $783.76 \pm 46.10$ | Ridge | TVAE | $0.528 \pm 0.063$ | $0.534 \pm 0.043$ |
| | | | | CTGAN | $0.471 \pm 0.080$ | $0.458 \pm 0.099$ |
| | | | | GC | $0.576 \pm 0.018$ | $0.579 \pm 0.017$ |
| | | | RF | TVAE | $0.626 \pm 0.028$ | $0.631 \pm 0.024$ |
| | | | | CTGAN | $0.576 \pm 0.020$ | $0.583 \pm 0.017$ |
| | | | | GC | $0.543 \pm 0.017$ | $0.534 \pm 0.011$ |
| | | | XGB | TVAE | $0.603 \pm 0.054$ | $0.606 \pm 0.040$ |
| | | | | CTGAN | $0.547 \pm 0.038$ | $0.559 \pm 0.019$ |
| | | | | GC | $0.543 \pm 0.017$ | $0.531 \pm 0.025$ |
| 2/3 | $46.67 \pm 1.11$ | $3021.49 \pm 478.42$ | Ridge | TVAE | $0.523 \pm 0.031$ | $0.464 \pm 0.130$ |
| | | | | CTGAN | $0.511 \pm 0.049$ | $0.412 \pm 0.078$ |
| | | | | GC | $0.576 \pm 0.018$ | $0.580 \pm 0.018$ |
| | | | RF | TVAE | $0.634 \pm 0.023$ | $0.578 \pm 0.094$ |
| | | | | CTGAN | $0.573 \pm 0.030$ | $0.557 \pm 0.038$ |
| | | | | GC | $0.543 \pm 0.017$ | $0.534 \pm 0.020$ |
| | | | XGB | TVAE | $0.615 \pm 0.042$ | $0.625 \pm 0.031$ |
| | | | | CTGAN | $0.549 \pm 0.036$ | $0.494 \pm 0.059$ |
| | | | | GC | $0.543 \pm 0.017$ | $0.533 \pm 0.033$ |
| 3/4 | $64.98 \pm 0.53$ | $8208.15 \pm 778.51$ | Ridge | TVAE | $0.464 \pm 0.009$ | $49.12 \pm 0.067$ |
| | | | | CTGAN | $0.548 \pm 0.044$ | $0.423 \pm 0.095$ |
| | | | | GC | $0.576 \pm 0.018$ | $0.582 \pm 0.018$ |
| | | | RF | TVAE | $0.6587 \pm 0.032$ | $0.5924 \pm 0.045$ |
| | | | | CTGAN | $0.578 \pm 0.028$ | $0.544 \pm 0.04$ |
| | | | | GC | $0.542 \pm 0.017$ | $0.537 \pm 0.016$ |
| | | | XGB | TVAE | $0.656 \pm 0.041$ | $0.0.643 \pm 0.031$ |
| | | | | CTGAN | $0.563 \pm 0.038$ | $0.515 \pm 0.029$ |
| | | | | GC | $0.542 \pm 0.017$ | $0.519 \pm 0.038$ |

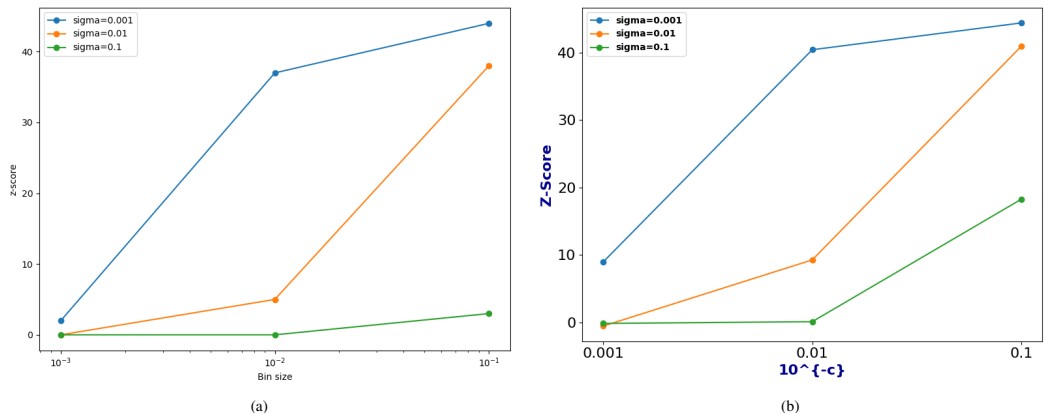

Figure 6: This figure shows the evaluation of the robustness of Gaussian noise by studying the z-score across various choices of standard deviation. To the left, we show the results from Ngo et al. (2024), and to the right, we show the results from our own experiment. Observe similar behavior across both works.

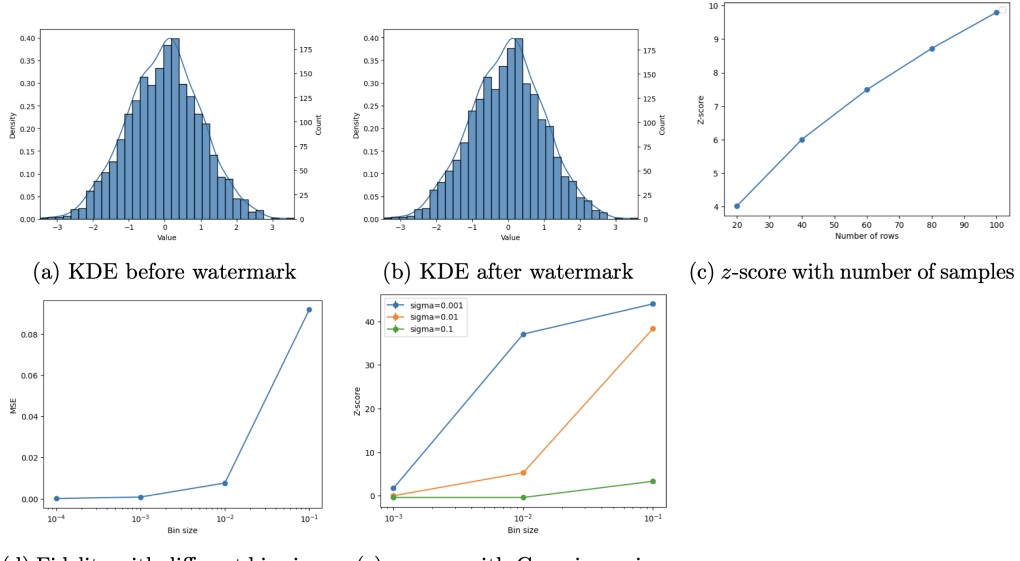

(a) KDE before watermark (b) KDE after watermark (c) $z$-score with number of samples

(d) Fidelity with different bin sizes (e) $z$-score with Gaussian noise

Figure 7: This is a reproduction of Figure 2 from Ngo et al. Ngo et al. (2024).

Recall that $\mathcal{H}$ maps to 0 and 1 with equal probability. Therefore, for a given $x'_i = x_i + k_i \cdot 10^{-c}$, the hash function should have mapped to 1 for every choice from 0 to $k_i - 1$ and succeed in time $k_i$. In other words, $\Pr[K_i = k] = \left(\frac{1}{2}\right)^{k+1}$, i.e., it follows a geometric distribution.

Now, $||\mathbf{X} - \mathbf{X}_w||_\infty = \max_i |x_i - x'_i| = \max_i k_i \cdot 10^{-c}$. We can use the well-known approximation for the maximum of $n$ i.i.d geometric variables to get $\mathbb{E}[\max_i k_i] = 0.5 + H_N / \ln 2$ where $H_N$ is the $N$-th harmonic number. Further $\ln N \le H_N \le 1 + \ln N$ or $H_N \le \ln N + 1$. This gives us that:

$$\mathbb{E}[||\mathbf{X} - \mathbf{X}_w||_\infty] \le \left(0.5 + \frac{\ln(N) + 1}{\ln 2}\right) \cdot 10^{-c}$$

$$\le (\ln N + 2) \cdot 10^{-c}$$

$\square$

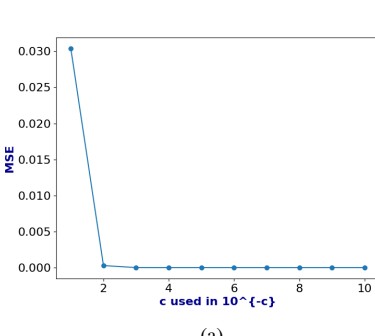 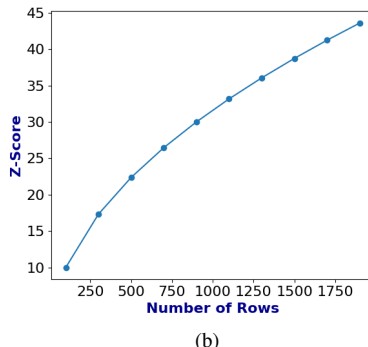

(a)             (b)

Figure 8: Plot of additional experiments on Gaussian dataset. Figure 8a plots MSE for more values of $c$. Figure 8b shows how the z-score changes when more rows are involved in the computation.

*Proof of Theorem 2.* The Jensen-Shannon Divergence (JSD) measures the similarity between two probability distributions. It is defined as:

$$JSD(P||Q) = \frac{1}{2}D(P||M) + \frac{1}{2}D(Q||M) \tag{2}$$

where $M = \frac{1}{2}(P+Q)$ is the midpoint distribution, and $D(P||Q)$ is the Kullback-Leibler Divergence, defined as: $D(P||Q) = \sum_x P(x)\log(\frac{P(x)}{Q(x)})$.

Let us find: $JSD(\rho||\rho')$. Partition the set of all values $X$ into $X_0$ and $X_1$ where $X_b$ consists of those values in $X$ that hashes to bit $b$. Note that $\rho'$ is only defined on $X_0$ giving:

$$\rho'(x) = \begin{cases} \frac{\rho(x)}{Z} & x \in X_0 \\ 0 & \text{otherwise} \end{cases}$$

Here, $Z$ is a normalization term needed to ensure that the sum of probabilities in $\rho'$ is 1. Since the hash function is ideal, i.e., maps to 0 and 1 with equal probability, $Z$ is approximately 0.5 or $\rho'(x) = 2 \cdot \rho(x)$ for $x \in X_0$.

Now, let's find the midpoint distribution $M(x) = \frac{1}{2}(\rho(x) + \rho'(x))$. We get:

$$M(x) = \begin{cases} \frac{3}{2}\rho(x) & x \in X_0 \\ \frac{1}{2}\rho(x) & \text{otherwise} \end{cases}$$

Now, we can compute the Kullback-Leibler divergences:

$$D(\rho||M) = \sum_{x \in X} \rho(x)\log(\frac{\rho(x)}{M(x)})$$

$$= \sum_{x \in X_0} \rho(x)\log(\frac{\rho(x)}{\frac{3}{2}\rho(x)}) + \sum_{x \in X_1} \rho(x)\log(\frac{\rho(x)}{\frac{1}{2}\rho(x)})$$

Simplifying, we get $D(\rho||M) = 0.5(\log(2) + \log(2/3)) = 0.5\log(4/3)$. Similarly, we get: $D(\rho'||M) = \log(4/3)$. Plugging this in Equation 2, we get:

$$JSD(\rho||\rho') = \frac{3}{4}\log(\frac{4}{3}) \approx 0.215$$

$\square$

*Proof of Proposition 1.* Of the $m$ values, we need to compute $T_i$ that ensures that the score is $\alpha$. We use Equation 1 as:

$$\alpha = \frac{2(T_i - 0.5M)}{\sqrt{M}}$$

Then, $T_i = 0.5M + \alpha\sqrt{M}/2$. In other words, we need at least $0.5M + \alpha\sqrt{M}/2$ values to ensure a Z-score of $\alpha$. Call this value $T_\alpha$. $\qquad\square$

*Proof of Theorem 3.* First, observe that for any value $val_i$ such that $\mathcal{H}(val_i) = 0$:

$$\Pr[\mathcal{H}(val_i + \epsilon_i) = 1] = \frac{1}{2}$$

for any $\epsilon_i \xleftarrow{\$} \mathcal{D}$. We already know that one needs at least $T_\alpha = 0.5M + \alpha\sqrt{M}/2$ cells to be unmodified to get a score of $\alpha$ (from Proposition 1). To achieve the watermark removal, we need to add noise to the remaining $N - T_\alpha$ cells. Observe that this follows a hypergeometric distribution - in a sample of size $M$, $N$ successes exist (i.e., mapping to 0). Then, the expected number of tries to pick at least $(N - T_\alpha)$ successfully is given by: $\approx (N - T_\alpha) \cdot M/N$. Therefore, we get:

$$\mathbb{E}[r] := 2 \cdot (N - T_\alpha) \cdot \frac{M}{N}$$

$\qquad\square$

