# OpenReview forum: "HashMark: Watermarking Tabular/Synthetic Data For Machine Learning Via Cryptographic Hash Functions"
_ICLR.cc/2026/Conference — Submitted to ICLR 2026_

### Official Review · Reviewer_VYef · 2025-10-24

**Soundness:** 2
**Presentation:** 3
**Contribution:** 1
**Rating:** 2
**Confidence:** 5

**Summary:**

This paper proposes a watermarking method for tabular data that modifies the data so that their hash values map to desired target bits. The idea is very similar to the red/green list watermark for LLMs. When inserting the watermark, they aim to make sure that each row (or cell) hash to the same target bit. Detection can then be conducted by counting the proportion of rows/cells whose hash values map to that target bit.

**Strengths:**

The paper is clearly written and easy to follow.

The detection algorithms are supported by mathematical theorems, though they rely on idealized assumptions about the hash function.

The algorithms are validated by various experiments.

**Weaknesses:**

1. When using the algorithm, the fidelity loss for the categorical data is too large. For example, in the HashMark_2, as indicated in the page 6 of the paper, all values are required to have the property of hashing to a desired bit. This essentially requires to modify about 50% of the data, which is a relatively large proportion.  Also, after applying the proposed algorithm, some of the categories will disappear. Therefore this algorithm seems not to be practical for categorical data.

2. For the numerical data, in the fidelity guarantee (theorem 1), we can see that there is a term $ln N$, where N is the size of the dataset. This seems to be strictly worser than the fidelity guarantee obtained in the previous work (https://arxiv.org/pdf/2405.14018), in which the fidelity guarantee does not rely on N.

3. Missing related works:
There are several other important recent tabular data watermark works this paper does not cite and discuss. For example:
https://openreview.net/forum?id=71pur4y8gs
https://openreview.net/pdf?id=3K4oAgZTcO
Please also cite and discuss the paper's advantage/disadvantage compared to these papers.

4. There are many errors in the theoretical part of this paper. For example, in the end of page 22,
$0.5+ \frac{\ln {N} +1}{ \ln {2}}$ is not smaller or equal to $\ln {N}+2$. In the proof of theorem 3, you use approximation when calculating the expected number of tries, however in the main theorem there is no $\approx$.

**Questions:**

The paper relies on a too idealized hash function assumption, saying any unwatermarked dataset can be split such that approximately half of the data are mapped to bit 0 and the other half to bit 1. This assumption seems to be very strong. It will be helpful to discuss in more detail from a cryptographic perspective, whether such an assumption makes sense or not.

---

> ### Author Response · Authors · 2025-12-03
> **A seeded hash function behaves like a pseudorandom function when the seed is unknown, this guarantees that hashing a dataset, whp, will guarantee a 50-50 split into 0 and 1. We also address other comments.**
>
> We thank the reviewer for raising this critical point. In our construction, the hash function is a seeded mapping, meaning it takes both a secret seed and the input as arguments. Under this formulation, the seeded hash function $H_s$​ can be viewed as a pseudorandom function (PRF), and its output is therefore indistinguishable from a uniform random value. This interpretation is supported by classical results showing that keyed hash constructions such as HMAC behave as PRFs under standard cryptographic assumptions (Bellare et al., 1996; Bellare and Rogaway, 1993; Goldreich et al., 1986). This modeling approach is also reflected in applied standards: NIST’s key-derivation functions explicitly rely on PRFs built from HMAC-style constructions (NIST, 2009), and modern deterministic random bit generators such as HMAC-DRBG used in production systems, including Linux, depend on the pseudorandomness of keyed hashing (NIST, 2015).
>
> We also address several additional reviewer comments below.
> - Response to Weakness (1): We appreciate the reviewer’s observation, but it actually highlights the core point of our paper. HashMark does not claim to solve watermarking for categorical data (i.e., ones with limited range); instead, we explicitly show why prior red–green (binary) hashing schemes are fundamentally incompatible with categorical attributes. As the reviewer notes, enforcing a target hash bit on categorical values would require altering roughly half of them, causing categories to collapse or disappear—exactly the limitation we critique in existing approaches such as TabularMark.
> HashMark’s goal is to make this difficulty clear, not to patch it. Instead, our proposed techniques intentionally focus on text-based, free-range features, where fine-grained numerical adjustments are meaningful and do not distort the underlying semantic distribution. In short, the reviewer’s concern is valid for categorical data in general, but it aligns with, rather than contradicts, the motivation and scope of our work.
> - Response to Weaknesses (2) and (4): We appreciate the reviewer identifying the typographical error in the simplification on page 22. The correct inequality is $0.5 + \frac{\ln N + 1}{\ln 2} \le \ln N + 3$ which is true for $N \ge 3$.
> However, we respectfully disagree with the interpretation that HashMark’s fidelity guarantee is “strictly worse” than prior work (e.g., arXiv:2405.14018). Theorem 1 in our paper depends on the dataset size $N$. Still, this dependence is only logarithmic and thus negligible in practice, especially when contrasted with the exponential suppression of error controlled by our parameter $c$. Prior work obtains an error bound of $ 1/m$ (the size of the bin), which is independent of $N$ but scales only linearly with $m$. HashMark’s error scales as $10^{−c}$. Even accounting for the highest-error outlier expected in a large dataset, HashMark achieves substantially higher fidelity. Let us consider $N=10^6$. Then we get the following values:
>   * HashMark: Setting $c=6$, we get: $\mathbb{E}[\vert X - X_w\vert_\infty]\le (1.4427 \cdot \ln(10^6) + 1.9427) \cdot 10^{-6} \approx \mathbf{0.00002187} $.
>   * Prior work of Ngo et al. (arXiv 2024): They use $m=1000$. The error is deterministic: $ ||X - X_w||_\infty \le \frac{1}{1000} = \mathbf{0.001} $.
>
> Thus, HashMark’s expected worst-case error is about 45 times smaller than the baseline error of Ngo et al. The mild logarithmic dependence on $N$ is far outweighed by the exponential improvement in precision from the parameter $c$.
>
> - Response to Weakness (3). We thank the reviewer for pointing us to TabWak: A Watermark for Tabular Diffusion Models (Zhu et al.) as relevant related work. We will cite and discuss it appropriately. TabWak presents a novel model-specific method that embeds watermarks into Gaussian latent codes of tabular diffusion models. However, TabWak faces a significant limitation: the substantially lower accuracy of Denoising Diffusion Implicit Models (DDIM) inverse processes in tabular diffusion models compared to image/video models. This occurs because tabular models incorporate additional algorithmic components, such as quantile normalization and Variational AutoEncoders (VAEs), that are difficult to reverse accurately. Since watermark detection requires inverting the entire data processing pipeline, these inversion errors accumulate, making watermark detectability highly dependent on model implementation and severely restricting practical utility (Fang et al., arXiv 2025). HashMark, by contrast, is a practical, model-agnostic, and scalable edit-based system that avoids inversion entirely. Detection requires only a simple cryptographic seed, and the method is naturally data-type agnostic, supporting both categorical and numerical attributes with flexible sparse or dense embedding. This makes HashMark significantly more efficient, robust, and broadly applicable for enterprise-level provenance and governance.

---

### Official Review · Reviewer_pZf5 · 2025-10-29

**Soundness:** 2
**Presentation:** 1
**Contribution:** 2
**Rating:** 2
**Confidence:** 4

**Summary:**

This paper proposes HashMark, a watermarking scheme for tabular data using cryptographic hash functions. HashMark₁ embeds pseudorandom bits in sparse locations, while HashMark₂ embeds a global bit across all cells using hash-based mapping and LSB perturbation (adding 10^-c) or rejection sampling. Detection uses cryptographic verification (HashMark₁) or statistical z-tests (HashMark₂). The authors claim advantages over recent work in simplicity, detection cost, and data-type support.

**Strengths:**

- Low detection cost: No model needed to embed the watermark
- Data-type flexibility: Hash functions naturally handle numeric and categorical data
- Simplicity: Easier to implement.

**Weaknesses:**

- Limited novelty: Hash watermarking, LSB modification, rejection sampling.. This is engineering refinement, not a research contribution.
- Misleading threat model: "Non-adversarial enterprise settings" assumption (lines 65-70) contradicts security claims throughout the paper. - If employees don't attack, why need cryptographic watermarks?
- Missing recent works such as TabWak and RINTAW.

**Questions:**

- What happens to robustness when data is rounded to 2-4 decimal places (common in practice)?
- “Zheng et al. (2024) requires the full source dataset.” What is the reason for this requirement?
- What is the performance of recent tabular diffusion models (such as TabDDPM, TabSyn, and TabDiff) when evaluated on both watermarked and non-watermarked data?

---

> ### Author Response · Authors · 2025-12-03
> **HashMark is a versatile, model-agnostic watermarking mechanism, bypassing the issues with detection in the case of TabWak (as it requires DDIM inversion) and scoring dependencies for RINTAW.**
>
> We thank the reviewer for their comments and questions regarding our work, particularly regarding prior literature. Below, we provide a detailed comparison by running additional experiments.
>
> - Both the choice of the parameter $c$ and the portion of the input consumed by the hash function (parameterized by $d$) are configurable. For example, assume the hash function processes only $d=4$ decimal places (and we set $c=4$ accordingly). If the original value is $x=1.742245$, the hash input becomes 1.7422. Any modification beyond the fourth decimal place, such as $x′=1.74225$ or $x′=1.7422$, will therefore not affect the input to the hash function, and the watermark will remain unchanged. As the reviewer correctly highlights, understanding how data will be represented or transformed in practice (e.g., via rounding, truncation, or quantization) is crucial when choosing dd and cc to ensure HashMark's robustness.
> - TabularMark embeds its signal in the difference between the source and destination datasets. For instance, if the original value is 14.5 and the green list contains 0.495, the modified value becomes 14.995. A drawback of this approach is that watermark detection requires access to the original value: one must compute the difference and check whether it belongs to the green list. HashMark avoids this dependency by embedding directly in the data or feature space without requiring access to the original dataset for verification.
> -  Experiments with Diffusion Models: We conducted further experiments using diffusion–model–based synthetic data generation to address the reviewer’s question. Using TabSyn on the *Shoppers* dataset, we observed the following model accuracy for the following classifiers:
>
>     *   XGBoost Classifier:
>            * Non-watermarked: \(89.60 \pm 0.34\)
>            * Watermarked: \(88.57 \pm 0.33\)
>    * Random Forest Classifier:
>           * Non-watermarked: \(90.24 \pm 0.39\)
>           * Watermarked: \(89.60 \pm 0.34\)
> Note that while synthetic datasets produced by diffusion models achieve higher accuracy than other methods, HashMark’s versatility is that it produces statistically insignificant changes in accuracy across all models.
> We also address additional reviewer comments below to resolve any misunderstandings.
> - Comment on Threat Model: Our assumption of non-adversarial enterprise settings, as in Ngo et al., reflects the goal of preventing accidental mismanagement and enabling verifiable provenance. Cryptographic watermarks ensure non-repudiation and high-fidelity detection, linking leaked datasets to their source even in unintentional leaks. The cryptographic design provides a robust, low-collision signal, enabling our main contribution: an exponentially strong fidelity guarantee of $10^{-c}$ with minimal perturbation, combining accountability with negligible utility loss.
> - Comment on Related Work: We have also revised the manuscript to expand the comparison with the works mentioned by the reviewer, which we summarize below.
>   * TabWak presents a novel model-specific method that embeds watermarks into Gaussian latent codes of tabular diffusion models. However, TabWak faces a significant limitation: the substantially lower accuracy of Denoising Diffusion Implicit Models (DDIM) inverse processes in tabular diffusion models compared to image/video models. This occurs because tabular models incorporate additional algorithmic components, such as quantile normalization and Variational AutoEncoders (VAEs), that are difficult to reverse accurately. Since watermark detection requires inverting the entire data processing pipeline, these inversion errors accumulate, making watermark detectability highly dependent on model implementation and severely restricting practical utility (Fang et al., arXiv 2025, Page 2). HashMark, by contrast, is a useful, model-agnostic, and scalable edit-based system that avoids inversion entirely. Detection requires only a simple cryptographic seed, and the method is naturally data-type agnostic, supporting both categorical and numerical attributes with flexible sparse or dense embedding. This makes HashMark significantly more efficient, robust, and broadly applicable for enterprise-level provenance and governance.
>   * RINTAW embeds its watermark via a Tournament Sampling approach, generating multiple candidate rows and selecting one based on a cryptographic hash key derived from specific column values. Detection requires not only the secret key but also knowledge of the precise mechanisms—namely, the score function and identification of the key columns—that link each row to the embedded watermark. Its reliance on column selection and scoring makes it computationally intensive and less flexible than more model-agnostic approaches like HashMark.
> We emphasize HashMark’s versatility: it supports watermarking both existing tabular data and data generated at sampling time, enabling detection without any extra computational overhead.

---

### Official Review · Reviewer_yUUp · 2025-10-30

**Soundness:** 2
**Presentation:** 2
**Contribution:** 2
**Rating:** 4
**Confidence:** 3

**Summary:**

This paper works on the problem of watermarking tabular data to ensure provenance and ownership, a task complicated by data heterogeneity and the low redundancy of tables. The authors introduce HashMark, a suite of watermarking protocols based on seeded cryptographic hash functions. Experimental results demonstrate that HashMark maintains high fidelity, with minimal impact on downstream machine learning model performance.

**Strengths:**

1. The method uses seeded hash functions for data-type agnostic watermarking. This is a simple yet effective approach applicable to numerical and alphanumeric data without complex data-specific transformations.
2. The paper provides formal fidelity guarantees. Theorem 1 bounds the expected L-infinity error for numerical perturbations, while Theorem 2 bounds the distributional shift for alphanumeric data using Jensen-Shannon Divergence.

**Weaknesses:**

1.  The description of Algorithm 1 is confusing, as it appears to mix elements of `HashMark1` (looping to `l`) and `HashMark2`, and its input parameters suggest it is designed for synthetic data generation. Please provide a separate, clear pseudocode for the embedding process of `HashMark2` on an **existing dataset**.
2.  The `Generate` function for alphanumeric data relies on rejection sampling from an underlying distribution `ρ`. The paper does not adequately explain how to obtain or approximate this distribution `ρ` when watermarking a **pre-existing, static tabular dataset**, which is a critical detail affecting the method's practicality and reproducibility.
3.  The proposed method of preserving correlations through row-wise rejection sampling is computationally expensive. A more in-depth quantitative analysis of the trade-off between the threshold `t`, computational cost, and the preservation of the joint data distribution beyond downstream task accuracy would strengthen the paper.

**Questions:**

1.  For watermarking a text column in an existing table (e.g., product descriptions), how do you propose to construct or estimate the probability distribution `ρ` for the `Generate` function's rejection sampling?

---

> ### Author Response · Authors · 2025-12-03
> **HashMark extended to now support free-range text data by using state-of-the-art embedding techniques.**
>
> We thank the reviewer for raising this question. Free-form text columns, such as product descriptions, do not admit a natural probability distribution for rejection sampling, and existing approaches like He et al., Ngo et al., or TabularMark, which rely on binary binning, distort semantics and can introduce artificial dataset skew.
>
> To address this, we can easily extend HashMark to support free-form text natively, without requiring randomization in its structure (as with ASINs or git-commit hashes) or bounded ranges, while preserving semantic structure. Our approach proceeds in two stages:
> - Stage 1: Semantic Embedding and Normalization: Each text entry is converted into a 384-dimensional embedding vector using a pre-trained language model, MiniLM-L6-v2. The final embedding vector is then L2-normalized to ensure a stable, robust semantic representation suitable for watermarking.
> - Stage 2: HashMark Watermarking: A small, deterministic perturbation ($10^{−6}$) is added to each embedding dimension, creating an imperceptible but globally correlated watermark that preserves semantic meaning and enables reliable detection. This stage is the same approach used for watermarking other numerical datasets, as stated in our existing experiments section.
>
> Experimental Evaluation: To evaluate robustness, we measure how consistently the watermark preserves semantic similarity in embedding space. In our experiments, we focused on the reviewer’s suggestion of using product descriptions. Specifically, we created a dataset of 1,000 base product descriptions and, for each base description, generated 20 semantically similar synthetic variations, resulting in 20,000 text pairs in total.
>
> For every pair consisting of an original product description A and one of its synthetic variations B, we computed:
> - the cosine similarity between the unwatermarked embeddings of A and B, and
> - the cosine similarity between the watermarked embeddings of A (say, A’) and B (Say, B’).
>
> We then measured the difference between these two similarity values to quantify the distortion introduced by the watermarking process. The difference in cosine similarity $(Cos(A,B)−Cos(A',B'))$ is the critical metric used to evaluate the imperceptibility and stability of the watermark in a semantic embedding space. We measure this difference between the original, unwatermarked semantic relationship $(Cos(A,B))$ and the watermarked relationship $(Cos(A',B'))$ for pairs of text entries. Across all 20,000 pairs, the average change in cosine similarity was $−1.91\cdot 10^{-8} \pm 7.2\cdot 10^{-7}$ (which is negligibly small). This quantifies how minimally the watermark has perturbed the semantic direction of the vectors. This proves that the watermarking process does not alter the relative meaning of the texts, ensuring that semantically similar documents remain highly similar and that semantic search results are unaffected by the presence of the hidden mark. In other words, the watermarking is semantically invisible.

---

### Official Review · Reviewer_vLiz · 2025-11-02

**Soundness:** 3
**Presentation:** 3
**Contribution:** 3
**Rating:** 6
**Confidence:** 3

**Summary:**

The paper “HashMark” presents a simple, hash-based framework for watermarking tabular and synthetic data to ensure provenance and ownership. Using seeded cryptographic hash functions, it embeds watermark bits with minimal distortion while preserving data utility. Two schemes are proposed: HashMark1 (sparse, high-fidelity) and HashMark2 (dense, statistically testable and type-agnostic). Theoretical bounds guarantee low perturbation, and experiments on multiple datasets show negligible (<0.5%) impact on ML accuracy. Compared with prior works such as Ngo (2024), He (2024), and Zheng (2024), HashMark offers broader data-type support, lower detection cost, and higher practicality for data governance.

**Strengths:**

1 Conceptual simplicity with wide scope: A seeded cryptographic hash as a unifying primitive for tabular watermarking is elegant and data-type agnostic.

2 Two complementary schemes: Sparse vs. dense addresses fidelity vs. robustness trade-offs.

3 Low-overhead detection: Moving from red/green binning to seed-hash + z-test lowers reliance on source data and reduces metadata burdens.

4 Practical angle: Results target non-adversarial enterprise governance (provenance auditing) rather than worst-case adversaries, which fits many real deployments

**Weaknesses:**

1 Adversarial robustness under-tested. While Theorem 3 and Proposition 1 give effort bounds, the empirical section lacks adaptive-attack evaluations (e.g., targeted noise, column selection, row injection crafted with knowledge of the scheme but not the seed). Robustness claims for HashMark2 rely on independence/CLT assumptions and ideal hashing; correlated columns or heavy-tailed noise could break z-test sensitivity. More thorough stress tests would strengthen soundness.


2 Categorical scope caveat. The paper recommends avoiding fixed-range categorical columns to prevent range gaps and distribution skew, which limits applicability in many tabular domains rich in discrete attributes.


3 Comparisons are partly qualitative. Head-to-head empirical comparisons to Zheng et al. (2024, TabularMark) and He et al. (2024) are minimal; the paper largely contrasts designs and reproduces Ngo et al. plots. A unified benchmark with common seeds/attacks would better substantiate superiority claims.


4 Correlation preservation is argued heuristically. The constrained-sampling strategy (threshold t) is practical, but correlation preservation is not theoretically guaranteed; guidance on selecting t versus expected z-score/utility would help. Runtime overheads grow rapidly at high t on some datasets.


5 Key/seed management not discussed. Deployment details (seed rotation, namespace collisions across datasets, auditing protocol) are not specified; these are important for provenance at scale.

**Questions:**

Q1 Can you provide adaptive-attack experiments (seed unknown) that optimize row additions/column removals/targeted noise to defeat the z-test while minimizing utility loss?

Q2 How sensitive is detection when columns are correlated or when hash inputs are truncated/normalized upstream (ETL effects)?

Q3 For fixed-range categorical columns (e.g., ICD codes), could you propose a lossless embedding variant (e.g., parity-preserving mapping, ECC over categories) instead of skipping them?

Q4 Could the z-test be extended to a likelihood-ratio test over all columns jointly to improve power under dependence?

Q5 What are operational protocols for seed management (per-dataset seeds, rotation, collision handling) and for public vs. private watermark verification?

---

> ### Author Response · Authors · 2025-12-03
> **HashMark is a simple model-agnostic watermarking scheme that simplifies prior designs.**
>
> We thank the reviewer for their feedback. We address the questions below in detail.
> - (Q1) With the seed unknown, a new cell hashes to 0 or 1 with roughly equal probability. This stems from prior work showing that $H_s$ can be viewed as a pseudorandom function, and that its output is therefore indistinguishable from that of a random function (Bellare, Canetti, & Krawczyk, 1996; Bellare & Rogaway, 1993; Goldreich, Goldwasser, and Micali, 1986). As a result, even duplicating all rows—as discussed in Lines 353–367—would lower the z-score while maintaining data utility. This reflects an inherent limitation of binning or binary-hashing–based watermarking methods and is not specific to HashMark.
> - (Q2) HashMark’s detection is intentionally sensitive to changes in the hash input—due to the use of a cryptographic hash—but is designed to be insensitive to column correlations and robust to small random perturbations common in ETL pipelines. Both the perturbation parameter $c$ and the hash input are tunable. For example, suppose the hash function only processes the first four decimal places of a numeric input. If the original value is $x=1.742245$, the hash will only consider 1.7422. Any truncation or rounding beyond the fourth decimal place, e.g., $x′=1.74225$ or $x′=1.7422$, will not affect the hash input, and thus the watermark remains unchanged. As the reviewer correctly notes, understanding how the data will be represented or manipulated in practice, such as rounding, truncation, or quantization, is critical when configuring HashMark to ensure robustness of the watermark.
> - (Q3)The goal of HashMark is to increase the number of elements mapping to 0 under the seeded hash. Any deterministic encoding risks skewing the dataset. For example, suppose the ICD code “I10” hashes to 1. If its nearest neighbor “I11” hashes to 0, then deterministically remapping all “I10” entries to “I11” would distort the distribution. A natural alternative is randomized embedding, but this requires storing the randomness used per row. A simpler way to realize randomized encoding is to allow additional inputs to the hash function beyond the categorical value—specifically, a per-row random salt. With a salt $r_i$​ for row $i$, the hash becomes $H(seed, ICD,r_i)$, so that, in expectation, half of the “I10” entries will map to 0. The drawback is that the salts must be stored for detection, which makes the dataset non-permutable and increases metadata overhead.
> - (Q4)Yes, the z-test can be extended to a multivariate likelihood-ratio test (LRT) that aggregates evidence across all columns. This is the statistically optimal approach for improving detection power, particularly when columns exhibit dependence. Our current method uses a column-wise z-test for simplicity and robustness, since it makes no assumptions about inter-column correlations. We will add a discussion of LRT-based multivariate detection for the threshold-based rejection-sampling scenario.
> - (Q5) Each dataset release must use a unique cryptographic seed $seed_{dataset}$ to ensure non-repudiation and limit the impact of any compromise. Cryptographically strong hash functions such as SHA-256 make collision handling unnecessary due to their negligible collision probability and strong signal integrity guarantees. HashMark currently supports private verification, since computing the watermark requires knowledge of $seed_{dataset}$. Designing a publicly verifiable variant is an interesting direction for future work.

---

### Author Response · Authors · 2025-12-03
**A Summary of the Discussion Phase**

We thank the reviewers for their feedback and insightful comments highlighting the strengths of our contribution:
- Simplicity and versatility: All reviewers noted HashMark's data-type agnostic hash-based approach
- Formal guarantees: Strong theoretical contributions via bounded $L_\infty$ error and Jensen-Shannon Divergence
- Clarity: Clear presentation, making the paper easy to follow
- Empirical validation: Extensive experiments supporting all stated claims

During the discussion phase, we provided the following additions:
- Free-form text handling: New experiments demonstrating HashMark's applicability to unrestricted text data using state-of-the-art text-to-vector embeddings (MiniLM-L6-v2), showing negligible semantic distortion across 20,000 product description pairs. This extends HashMark's versatility beyond randomized alphanumeric values.
- Diffusion model evaluation: New experiments using diffusion-based generators (TabSyn), reporting classification accuracy under watermarked vs. non-watermarked training data. Results show changes remain statistically insignificant, even for diffusion-based models.
- Expanded comparisons: Added detailed comparisons with TabWak and RINTAW, explaining their limitations. TabWak's detection requires DDIM inversion with low accuracy for tabular data, while RINTAW's embedding and detection require reproducing the scoring algorithm and identifying columns for each row. These comparisons reinforce why HashMark's simplicity and robustness make it a superior alternative.

We have improved clarity, expanded related work, and incorporated additional experiments to address all reviewer comments and questions fully. These revisions significantly strengthen the paper's contributions.

---

### Meta-Review · Area_Chair_6qWF · 2025-12-15

**Summary:**

Given the lack of experiments, limited novelty, practicality of the algorithm and the errors in the theoretical analysis, AC believes major concerns remain even after the rebuttal (see my comments below). Most reviewers vote for rejection and no reviewer champions the paper. Therefore, AC would recommend rejection of this paper too.

**Reviewer Concerns:**

Reviewer concerns that AC thinks were addressed:

1. Misleading threat model.

AC's comment: AC believes the rebuttal has clarified the motivation of the threat model.

2. Missing recent works such as TabWak and RINTAW.

AC's comment: The authors have addressed the missing references in the rebuttal.

Reviewer concerns that are still outstanding:

1. Adaptive-attack experiments.

AC's comment: The authors didn't provide an additional experiments to show the performance of the proposed method under adaptive attacks.

2. Limited novelty.

AC's comment: The authors didn't respond to this comment directly.

3. This algorithm seems not to be practical for categorical data.

AC's comment: In the rebuttal, the authors admitted that HashMark’s goal is to make this difficulty clear, not to patch it.

4. There are many errors in the theoretical part of this paper.

AC's comment: Though fixed, the authors admitted the error in the rebuttal. This lowers the credibility of the paper.

**Reviewer Scores:**

Many big concerns remain, such as the lack of experiments, limited novelty, practicality of the algorithm, and the errors in the theoretical analysis. The current version is not ready to appear in ICLR. AC believes the reviewers will keep their scores after the rebuttal.

---

### Decision · Program_Chairs · 2026-01-26

Reject